# Earthquake swarms frozen in an exhumed hydrothermal system (Bolfin Fault Zone, Chile)

Simone Masoch[1+], Giorgio Pennacchioni[1], Michele Fondriest[1], Rodrigo Gomila[1], Piero Poli[1], José Cembrano[2,3], Giulio Di Toro[1,4]

[1] Dipartimento di Geoscienze, Università degli Studi di Padova, Padua, Italy
[2] Departamento de Ingeniería Estructural y Geotécnica, Pontificia Universidad Católica de Chile, Santiago, Chile
[3] Andean Geothermal Center of Excellence (CEGA, FONDAP-CONICYT), Santiago, Chile
[4] Sezione di Roma 1, Istituto Nazionale di Geofisica e Vulcanologia, Rome, Italy
[+] Present address: Nevada Seismological Laboratory, University of Nevada, Reno, USA

*Correspondence to*: Simone Masoch (simone.masoch@unipd.it ; smasoch@unr.edu)

**Abstract.** Earthquake swarms commonly occur in upper-crustal hydrothermal-magmatic systems and activate mesh-like fault networks. How these networks develop through space and time along seismic faults is poorly constrained in the geological record. Here, we describe a spatially dense array of small-displacement (< 1.5 m) epidote-rich fault-veins (i.e. hybrid extensional-shear veins) within granitoids, occurring at the intersections of subsidiary faults with the exhumed seismogenic Bolfin Fault Zone (Atacama Fault System, Northern Chile). Epidote hybrid extensional-shear veining occurred at 3-7 km depth and 200-300 °C ambient temperature. At distance ≤ 1 cm to fault-veins, the magmatic quartz of the wall-rock shows (i) thin (< 10-µm-thick) interlaced deformation lamellae, and (ii) systematically crosscutting veinlets healed by quartz and feldspars, and appears shattered at the vein contact. Clasts of deformed magmatic quartz, with deformation lamellae and healed veinlets, are included in the epidote-rich fault-veins. Deformation of the wall-rock quartz is interpreted to record the transient large stress perturbation associated with the propagation of small earthquakes, preceding conspicuous epidote mineralization. Conversely, the epidote-rich fault-veins record cyclic events of extensional-to-hybrid veining and either aseismic or seismic shearing. The dilation and shearing behavior of the epidote-rich fault-veins are interpreted to record the later development of a mature and hydraulically-connected fault-fracture system. In this latter stage, the fault-fracture system cyclically ruptured due to fluid pressure fluctuations, possibly correlated with swarm-like earthquake sequences.

## 1 Introduction

The thermo-hydro-mechanical and chemical properties of fault zones and their host rocks affect a wide range of processes in the Earth's crust, such as earthquake nucleation, propagation and arrest (e.g., Faulkner et al., 2006; Sibson, 1985; Wesnousky, 1988, 2006), crustal rheology (e.g., Behr and Platt, 2014; Duan et al., 2024; Handy et al., 2007) and migration of fluids (e.g., hydrothermal, magmatic, oil, gas: Cembrano and Lara, 2009; Mittempergher et al., 2014; Richards, 2013; Tardani et al., 2016). The mechanical and hydraulic proprieties of fault zones vary largely through space and time during the seismic cycle and are intrinsically coupled (Caine et al., 1996; Faulkner et al., 2010; Wibberley et al., 2008). In particular, permeability changes during the seismic cycle at seismogenic depths are expected to promote co- to post-seismic episodic fluid flow (i.e. fault-valve behavior: Sibson, 1992a, 1992b, 1989). Indeed, fault rupture events can lead to large, transitory increases of fault permeability (Cox, 2016; Sibson, 1989). Where ruptures breach overpressured fluid reservoirs, high-permeability fault segments provide conduits facilitating fluid redistribution in the Earth's crust. On the other hand, post- to inter-seismic fault healing and sealing due to compaction and precipitation of hydrothermal minerals in pores and fractures reduce fault permeability, eventually arresting fluid flow (Cox, 2016; Sibson, 1992b, 1992a, 1989).

The expression of the coupling among fault activity, fault permeability, fluid flow, fluid pressure and loading conditions in the geological record is documented by hydrothermal (e.g., epidote, quartz, chlorite, calcite, zeolite) fault-vein networks in exhumed fault zones over several geological settings (e.g., Akker et al., 2023; Cerchiari et al., 2020; Cox and Munroe, 2016; Dempsey et al., 2014; Lucca et al., 2019; Malatesta et al., 2021; Masoch et al., 2022; Micklethwaite et al., 2010; Molli et al., 2010; Nüchter and Stöckhert, 2008, 2007; Ujiie et al., 2018). Mineralized fault-fracture networks display extensive hydrothermal alteration, mutually overprinting extension-to-hybrid vein arrays and dilatant breccias (Cox, 2016; Sibson, 2020). These features record significant stages of fluid flow and mineral precipitation during fault evolution, possibly associated with ancient seismic activity (e.g., Boullier and Robert, 1992; Cox, 2020; Cox and Munroe, 2016; Dempsey et al., 2014; Fagereng et al., 2010; Genna et al., 1996; Giuntoli and Viola, 2021; Micklethwaite and Cox, 2004; Muñoz-Montecinos et al., 2020; Ujiie et al., 2018). In recently or currently active hydrothermal-magmatic settings, abundant fluid flow is commonly accompanied by earthquake swarms (e.g., Danré et al., 2022a; Enescu et al., 2009; Fischer et al., 2014; Legrand et al., 2011; Mesimeri et al., 2021; Passarelli et al., 2018; Shelly et al., 2016, 2013; Yukutake et al., 2011), i.e. clusters of low magnitude seismic events without a characteristic mainshock (Mogi, 1963). Earthquake swarm events, lasting from a few days to months (e.g., Fischer et al., 2014), are driven by either pore fluid pressure fluctuations (e.g., Baques et al., 2023; Hill, 1977; Ross and Cochran, 2021; Shelly et al., 2022; Sibson, 1996) and aseismic slip (e.g., Danré et al., 2022b; De Barros et al., 2020; Lohman and McGuire, 2007; Vidale and Shearer, 2006). Besides deviating from common mainshock-aftershock sequences, earthquake swarms generate also considerable non-double-couple (i.e. isotropic) seismic signal, as a result of tensile fracturing and hybrid faulting attributed to the ingression of pressurized fluids in the fault zone/system (Legrand et al., 2011; Phillips, 1972; Sibson, 1996; Stierle et al., 2014; Vavryčuk, 2002). Similar human-induced seismic sequences may be associated with industrial fluid injection in boreholes (e.g., Ellsworth, 2013; Goebel et al., 2016; Guglielmi et al., 2015; Healy et al., 1968).

There has been a great deal of progress in the last years regarding (i) the imaging of fault networks illuminated by earthquake swarms (e.g., Baques et al., 2023; Ross et al., 2020; Shelly et al., 2022), (ii) the determination of focal mechanisms of very small-in-magnitude earthquakes through seismological analysis (e.g., Essing and Poli, 2022; Mesimeri et al., 2021; Poli et al., 2021), and (iii) the relation of injected fluid volumes and rates with seismic energy release through fluid-injection experiments (e.g., Dorbath et al., 2009; Guglielmi et al., 2015; McGarr, 2014). Many authors proposed that swarm-like earthquake sequences activate km-scale mesh-like fault-fracture networks in zones of fault geometric complexity, such as fault linkages and step-overs (e.g., Hill, 1977; Ross et al., 2020, 2017; Shelly et al., 2022, 2015; Sibson, 1996; Sykes, 1978). However, to date, how a fault-fracture network develops both in space and time in seismically-active hydrothermal systems is poorly constrained due to (i) the poor spatial resolution (> 10s of meters) of seismological and geophysical techniques relative to the length of (micro-)fracture processes and (ii) the limited exposure at the Earth's surface of exhumed fault-vein networks large enough to be comparable to currently active cases.

In this work, we examine the microstructures of an extensive epidote-rich fault-vein network located at a zone of fault linkage of the Bolfin Fault Zone (BFZ), an exhumed, crustal-scale, seismogenic (pseudotachylyte-bearing) fault of the transtensional Coloso Duplex (Atacama Fault System, Northern Chile, Fig. 1) (Cembrano et al., 2005; Masoch et al., 2022, 2021; Scheuber and González, 1999). The selected extensive epidote-rich fault-vein networks are well-exposed at centimeter-to-decameter scales over tens of square kilometers in the Atacama Desert (Fig. 1a) and have been proposed to represent an ancient upper-crustal seismically-active hydrothermal system, possibly capable to have produced swarm-like earthquake sequences (Masoch et al., 2022). Specifically, in this contribution, we aim at assessing the deformation processes governing the development of a potential upper-crustal swarmogenic volume. We document that the proximal wall-rock of small-displacement (< 1.5 m) fault-veins initially experienced a large transient stress pulse, attested by the occurrence of deformation lamellae within magmatic quartz. This deformed quartz is

included as clasts within epidote-rich fault-veins, that record overprinting events of extensional veining and cataclasis. We interpret these microstructures as evidence of ancient swarm-like activity, from the incipient stages of dynamic crack propagation to the later cyclic crack opening and shearing, driven by fluid pressure fluctuations, within a mature and hydraulically connected fault-fracture system. These exposed fault-vein networks represent a unique geological record of the evolution in space and time of a potential upper-crustal swarm-like seismic source, from the incipient stages of the propagation of a newly-produced micro-fracture network to the later development of a mature fault system.

## 2 Geological setting

The >40-km-long BFZ pertains to the 1000-km-long, Early Cretaceous, strike-slip intra-arc Atacama Fault System (Northern Chile; Fig. 1) (Arabasz, 1971; Cembrano et al., 2005; Masoch et al., 2021; Scheuber and González, 1999; Seymour et al., 2021). The BFZ displays sinistral strike-slip kinematics and bounds the western side of the crustal-scale transtensional Coloso Duplex (Cembrano et al., 2005; Masoch et al., 2022, 2021) (Fig. 1a). At regional scale, the BFZ has a sinuous geometry across Jurassic-Early Cretaceous diorite-gabbro and tonalite-granodiorite plutons (Fig. 1a). The ancient (125-118 Ma) BFZ seismicity is attested by presence of pseudotachylytes, formed at 5-7 km depth and $\leq 300$ °C ambient temperature (Gomila et al., 2021; Masoch et al., 2022, 2021). Seismic faulting occurred in a fluid-rich environment as documented by syn-kinematic chlorite-epidote (-quartz-calcite) veining and extensive propylitic alteration (Gomila et al. 2021).

In detail, the BFZ architecture consists of multiple (ultra)cataclastic strands, up 6-m-thick, within a 150-m-wide damage zone (see Masoch et al., 2022 for the description of the fault architecture; Fig. 1b). The damage zone consists of variably fractured and brecciated rock volumes characterized by extensive epidote-rich mesh-like fault-vein networks associated with NW-to-WNW-striking faults splaying from the BFZ (Figs. 1b-c; 2) (Masoch et al., 2022). These subsidiary faults accommodated transtensional slip (Fig. 1c) within the Coloso Duplex (Cembrano et al., 2005; Veloso et al., 2015), with an apparent cumulative strike-slip displacement up to 1 km (Cembrano et al., 2005; Jensen et al., 2011; Stanton-Yonge et al., 2020). The epidote-rich fault-vein networks consist of (i) hybrid extensional-shear veins (i.e. fault-veins) with lineated slickensides (Fig. 2a, 2c), and (ii) extensional and hybrid veins, and dilatant breccias, including fragments of altered wall-rocks and earlier veins sealed by epidote + prehnite ± chlorite ± quartz ± K-feldspar (Fig. 2b, 2d; see section 4.2). The fault-veins extend up to tens of meters in length (Figs. 1b, 2a), accommodated a cumulative displacement up to 1.5 m (Figs. 1b, 2a), are arranged in four sets, dipping towards SW, NE, NW and S (Fig. 1c), and are surrounded by extensive reddish alteration haloes in the damaged wall-rock (Fig. 2), as the small-displacement faults by Faulkner et al. (2011) . Epidote lineated slickensides are decorated by either stepped polished surfaces or mirror-like slip surfaces (Fig. 2a, 2c), and their kinematics range from normal dip-slip to strike-slip (either sinistral and dextral; Fig. 1c). The epidote-rich fault-vein networks observed in the BFZ damage zone are spatially distributed within all the duplex (Fig. 1a) and formed at 3-7 km depth and 200-300 °C (Herrera et al., 2005; Masoch et al., 2022; Olivares et al., 2010).

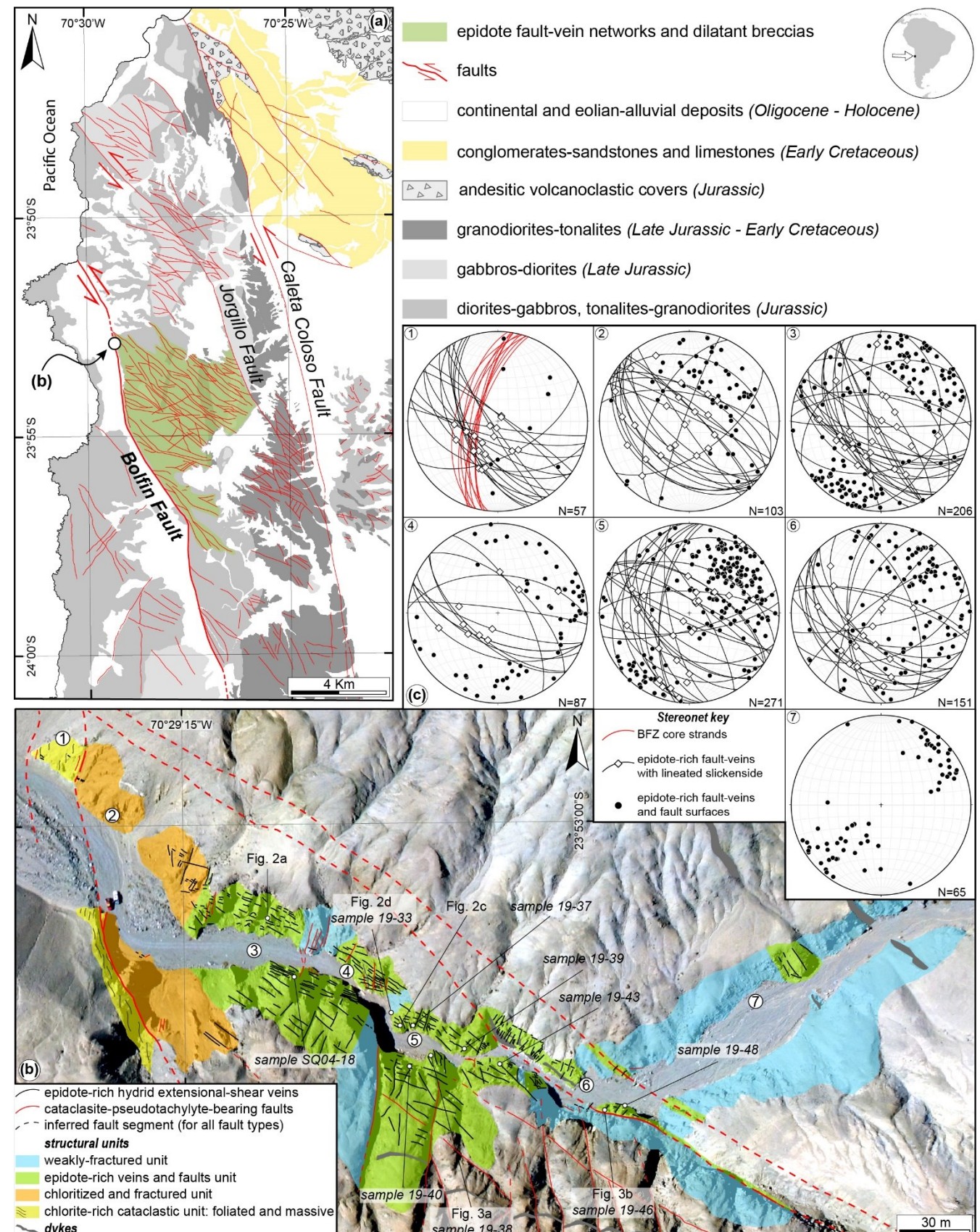

**Figure 1. Geological setting of the Bolfin Fault Zone. (a) Simplified geological map of the Coloso Duplex. The BFZ bounds the western side of the crustal-scale transtensional duplex. The green area indicates the distribution of the epidote-rich fault-vein networks and dilatant breccias within the Coloso Duplex. Modified from Cembrano et al. (2005). (b) Structural map of the BFZ architecture at Sand Quarry locality. Clusters of epidote-rich fault-vein networks and breccias are associated with NW-striking, splay faults of the BFZ, and NE-striking faults. The faults splaying out from the BFZ represent transtensional faults within the duplex (thick red lines). Modified**

from Masoch et al. (2022). (c) Structural data of the fault core strands and epidote-rich fault-vein networks. Numbers in stereonets
denote the location of structural sites in the map in (b).

## 3 Samples and methods

Nine samples representative of the different types of faults and veins pertaining to the epidote-rich fault-vein network were studied: (i) hybrid extensional-shear veins (i.e. fault-veins) with both wall-rock sides preserved (samples SQ04-18 and 19-48), (ii) fault-veins with only the footwall wall-rock preserved and lineated polished slickenside (samples 19-33, 19-38, 19-39 and 19-46), and fault-veins with only the footwall wall-rock preserved and lineated steeped slickenside (samples 19-37, 19-40 and 19-43). Microstructural analysis was conducted on Syton-polished 30- and 100-µm-thick thin sections (n=10) cut parallel to the fault lineation ($X$ direction) and orthogonal to the fault/vein wall ($X$-$Y$ plane). 100-µm-thick thin sections were produced to preserve the slickenside, where only the footwall wall-rock was present. Transmitted-light microscopy (OM) was used to determine microstructural features at thin-section scale and to identify areas suitable for electron microscopy investigations. We used a Tescan Solaris (Field Emission Gun – Scanning Electron Microscope; FEG-SEM) installed at the Department of Geosciences of University of Padova (Italy). The instrument is equipped with backscattered electron (BSE), cathodoluminescence (CL; wavelength detection range: 350-600 nm), electron backscattered diffraction (EBSD), and quantitative wavelength-dispersive spectroscopy (WDS) detectors. BSE and CL images were acquired at 5-10 kV and 0.3-3 nA, and 10 kV and 1-3 nA as accelerating voltage and beam current, respectively. The EBSD maps were acquired using the FEG-SEM equipped with a CMOS-Symmetry EBSD detector (AZtec acquisition software, Oxford Instruments), operating at 20 kV as accelerating voltage, 5-10 nA as beam current, 0.15-0.30 µm as step size, 70° sample tilt and high vacuum. Noise reduction was performed using the software CHANNEL5 of HKL Technology, Oxford Instruments, by removing wild spikes (i.e. single pixels surrounded by 8 neighbors with different orientations) and replacing zero-solution points with the orientation of nearest neighbors starting from eight neighbors down to five. EBSD data were processed using the MTEX toolbox (https://mtex-toolbox.github.io/).

The composition of main mineral phases was obtained by WDS-FEG analysis. Acquisition conditions were: 15 kV (accelerating voltage); 6 nA (beam current); 1 µm (electron beam size); 5 s (counting time for background), 15 s (for Si, Al, Ca, Fe), and 10 s (for Na, K, Mg, Mn, Ti, Cr) on peak. Albite (Si, Al and Na), diopside (Ca), olivine San Carlos (Mg), orthoclase (K), hematite (Fe), and Cr, Ti and Mn oxides were used as standards. Na and K were analyzed first to prevent alkali devolatilization affects.

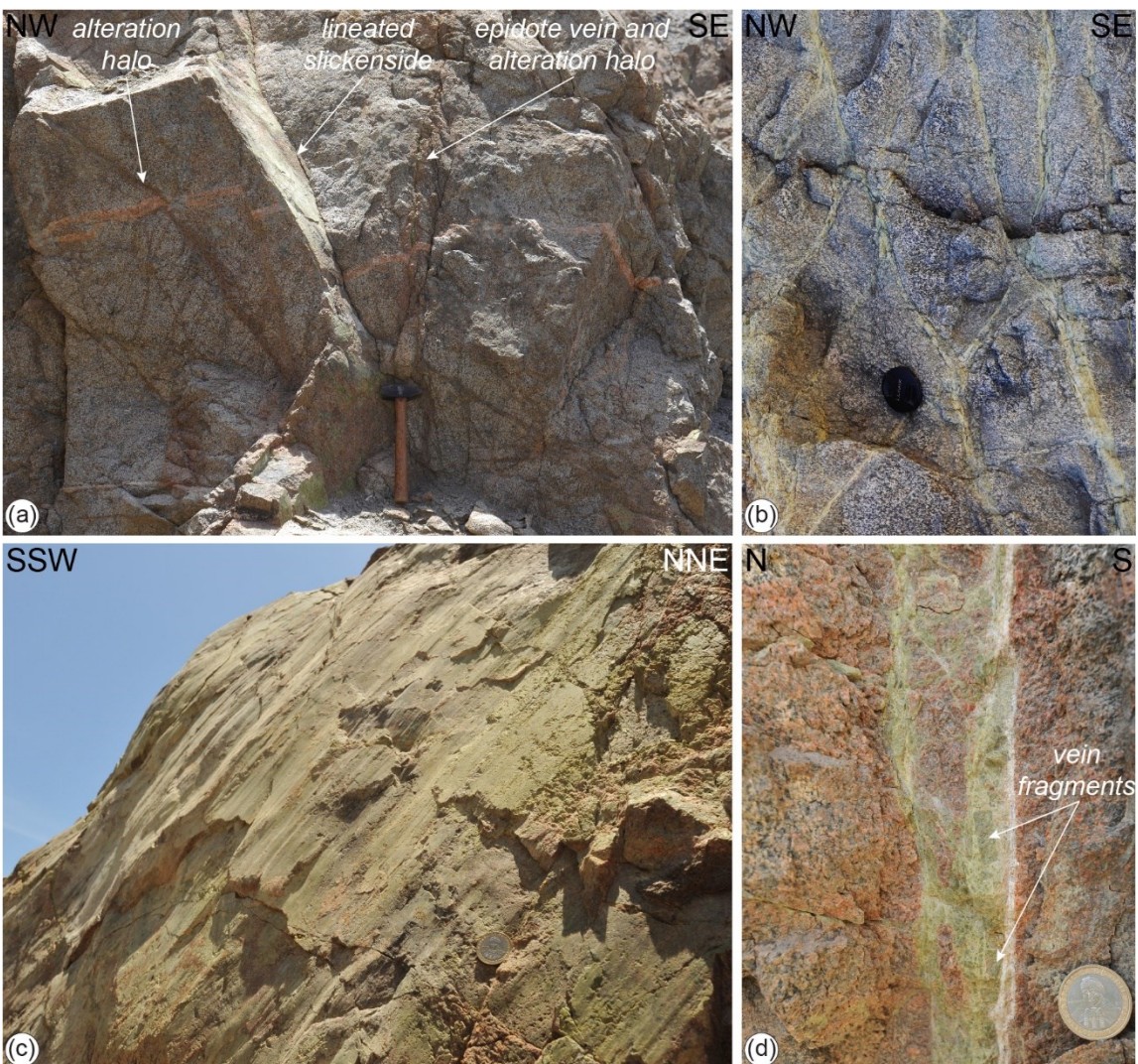

**Figure 2. The epidote-rich mesh-like fault-vein network of BFZ. Coin, hammer and cover lens for scale. (a) Hybrid extensional-shear veins and veins are surrounded by a red alteration halo in the damaged wall-rock. Lineated slickensides displace an aplitic dyke accommodating up to tens of centimeters of displacement. WGS GPS location: 23.44368°S, 70.487104°W. (b) Honeycomb mesh structure. WGS GPS location: 23.934255°S, 70.465309°W. Modified from Masoch et al. (2022). (c) Discrete extensional fault surface decorated by epidote slickenfibers. WGS84 GPS location: 23.883944°S, 70.486689°W. Modified from Masoch et al. (2022). (d) Epidote-rich fault-vein including angular fragments of earlier veins (dark green). The fault-vein is reactivated by a whitish calcite-palygorskite vein (boundary on the right side), referable to post-Miocene deformation (see Masoch et al., 2021 for details). Sample 19-33. WGS GPS location: 23.99803°S, 70.44051°W. Modified from Masoch et al. (2022).**

## 4 Results

### 4.1 Weakly-deformed granodiorite and micro-damage zone of the fault-veins

The weakly-deformed granodiorite consists of plagioclase (labradorite to andesine; Masoch et al., 2022), quartz, K-feldspar with myrmekite, biotite, minor amphibole, ilmenite and magnetite (Fig. 3a-c). The magmatic quartz shows weak undulose extinction (Fig. 3c) and has a dominant bright to light grey CL shade locally cut by CL-dark micro-fractures (>10 µm in thickness) sealed by hydrothermal quartz ± K-feldspar (Fig. 4a-b).

The granodiorite adjacent to epidote-rich fault-veins is turned into reddish alteration haloes, up to 4 cm in thickness (Figs. 2a, 2c-d, 3a-c), associated with (i) replacement of magmatic plagioclase by albite + epidote, and of magmatic biotite and amphibole by chlorite ± opaques (Fig. 3a-c), (ii) pervasive micro-fracturing, filled with epidote ± chlorite ± prehnite (Fig. 3b-c), and (iii) deformation of the magmatic quartz (Figs. 3c-e, 4-5, S1-S3). Quartz deformation microstructures include (1) deformation lamellae

(see EBSD data description below) (Figs. 3d-e, S1a-d), up to 10-µm-thick, visible in CL by the darker shade crosscutting the bright to medium grey-shaded host quartz (Figs. 4c-h, S1d, S2, S3a). The deformation lamellae appear straight under the optical microscope (Figs. 3d-e, S1a-b) and in the CL images (Figs. 4d, S1d, S2-S3a) and become interlaced and wavy when approaching the vein boundaries (Fig. 4h). The quartz deformation lamellae are systematically crosscut by (2) thin (up to 15-µm-thick) micro-fractures (hereafter referred as healed veinlets or veinlets) healed by quartz ± K-feldspar, across quartz grains (Fig. 4e-f), and K-feldspar ± albite when extending across neighbor feldspar grains (Fig. 4i-j). These veinlets are outlined by fluid inclusion trails, across magmatic quartz grains, under the optical microscope (Figs. 3c-e, S1), show a homogeneous dark (i.e. black) CL shade across host quartz and feldspar grains (Figs. 4d, 4f, 4h, 4j, S1d, S2-S3a) and are oriented at high angle with respect to the vein boundary (Figs. 3d, 4h). These deformation microstructures, i.e. quartz deformation lamellae and healed veinlets (hereafter referred to as "micro-damage zone"), fade away from the fault-veins and disappear at distances ≥ 1 cm (Fig. 4a-b). In the micro-damage zone, the healed veinlets increase in spatial density towards the fault-veins (Figs. 3c-e, 4c-j, S1-S2), while no apparent change in density of the quartz deformation lamellae is observed. In the footwall block, at < 100 µm distance from the sharp vein boundary, the magmatic quartz is strongly brecciated and healed by CL-dark grey-shaded quartz (also surrounded by epitaxial rim of CL-dark quartz; Fig. 4i-j).

EBSD maps of the quartz show that the deformation lamellae visible in CL are oriented nearly orthogonal to the <c> axis (i.e. sub-parallel to the basal plane; Figs. 5a-b, S1e-f, S3b) and correspond to a minor crystallographic misorientation (< 2-3°; see profiles in Fig. 5c-d, S1g, S3c) with respect to the host grain, which are the typical features of deformation lamellae in quartz (Carter, 1965; Christie et al., 1964; Drury, 1993; Fairbairn, 1941; McLaren et al., 1970; Trepmann and Stöckhert, 2003; White, 1973). . The EBSD maps also show that and the healed veinlets overgrew in epitaxial continuity with the host magmatic quartz (Figs. 5a, S1e).

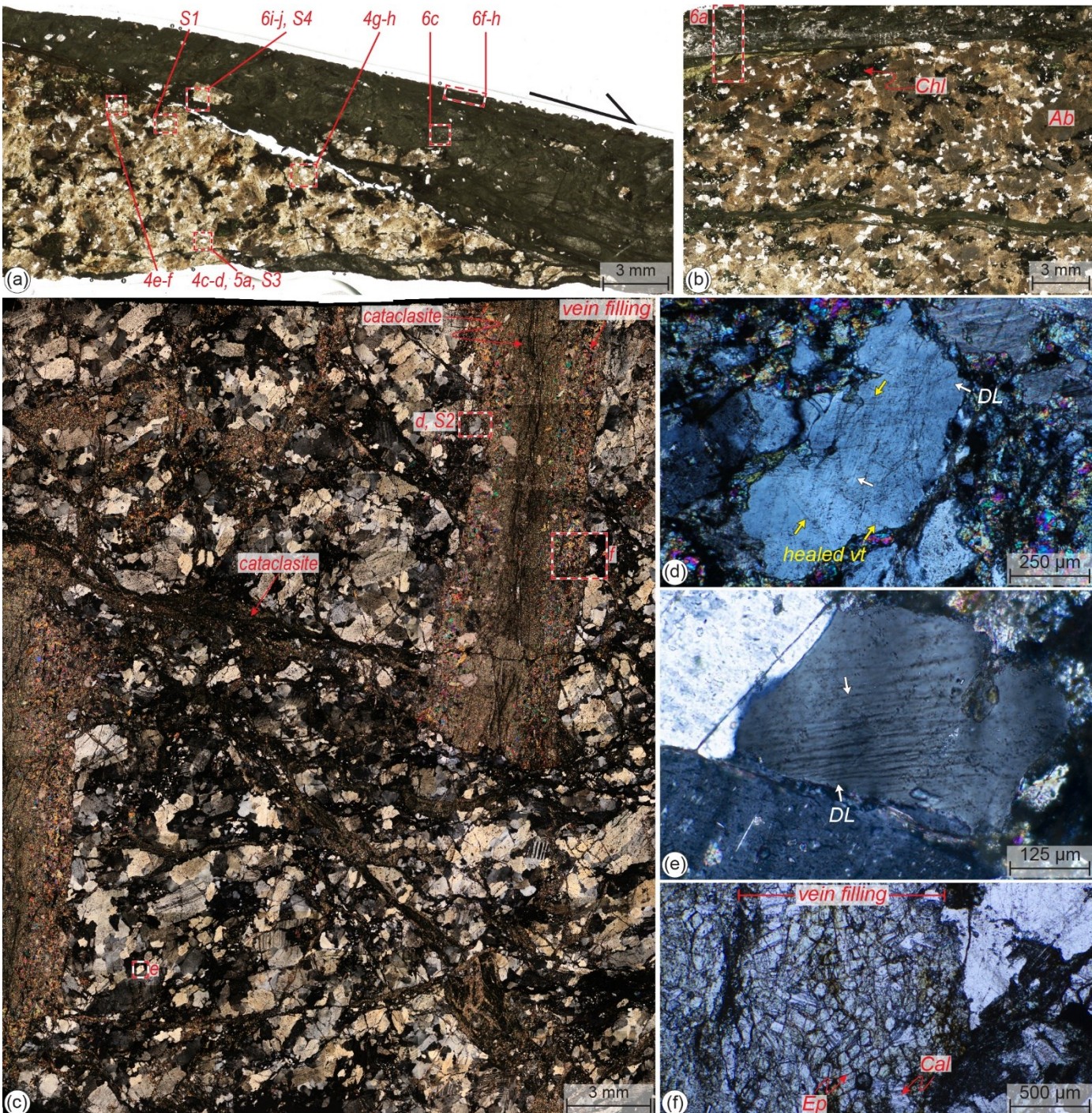

**Figure 3. Microstructures of the fault-veins and their associated wall-rock under the optical microscope. Mineral abbreviations:** *Ab* = albite, *Cal* = calcite, *Chl* = chlorite, *Ep* = epidote. (a) Plane-polarized light scan of thin section of a lineated fault-vein, showing the spatial distribution of the microstructures observed in the micro-damage zone and in the vein (dashed red boxes). Sample 19-38. (b) Plane-polarized light scan of thin section of a fault-vein recording multiple episodes of extensional-to-hybrid veining and along vein-boundary shearing. Sample 19-46. WGS84 GPS location 23.88428°S, 70.48615°W. The dashed rex box marks the zoom shown in Fig. 6a. (c) Cross-polarized light thin section micrograph of an extensional (*vein filling*) to shear (*cataclasite*) vein displaced by an epidote cataclasite. Quartz grains show undulose extinction in the weakly deformed granodiorite; while, they exhibit deformation lamellae and a dense pattern of fluid inclusion trails in the micro-damage zone. Dashed red boxes marks the zooms shown in (d-f). Sample SQ04-18. WGS84 GPS location 23.883906 °S, 70.486942 °W. (d) Quartz grain with deformation lamellae (white arrows; *DL*) cut by a dense pattern healed micro-fractures (*healed vt*) outlined by fluid inclusion trails (yellow arrows), whose most pervasive set is oriented perpendicular to the vein boundary. Cross-polarized light micrograph. The corresponding CL image is shown in Fig. S2 (e) Quartz grain with straight and narrow deformation lamellae (white arrows). Cross-polarized light micrograph. (f) Idiomorphic epidote and minor calcite crystals in the outer part of the vein. The inner part consists of a fine-grained cataclasite. Plane-polarized light micrograph.

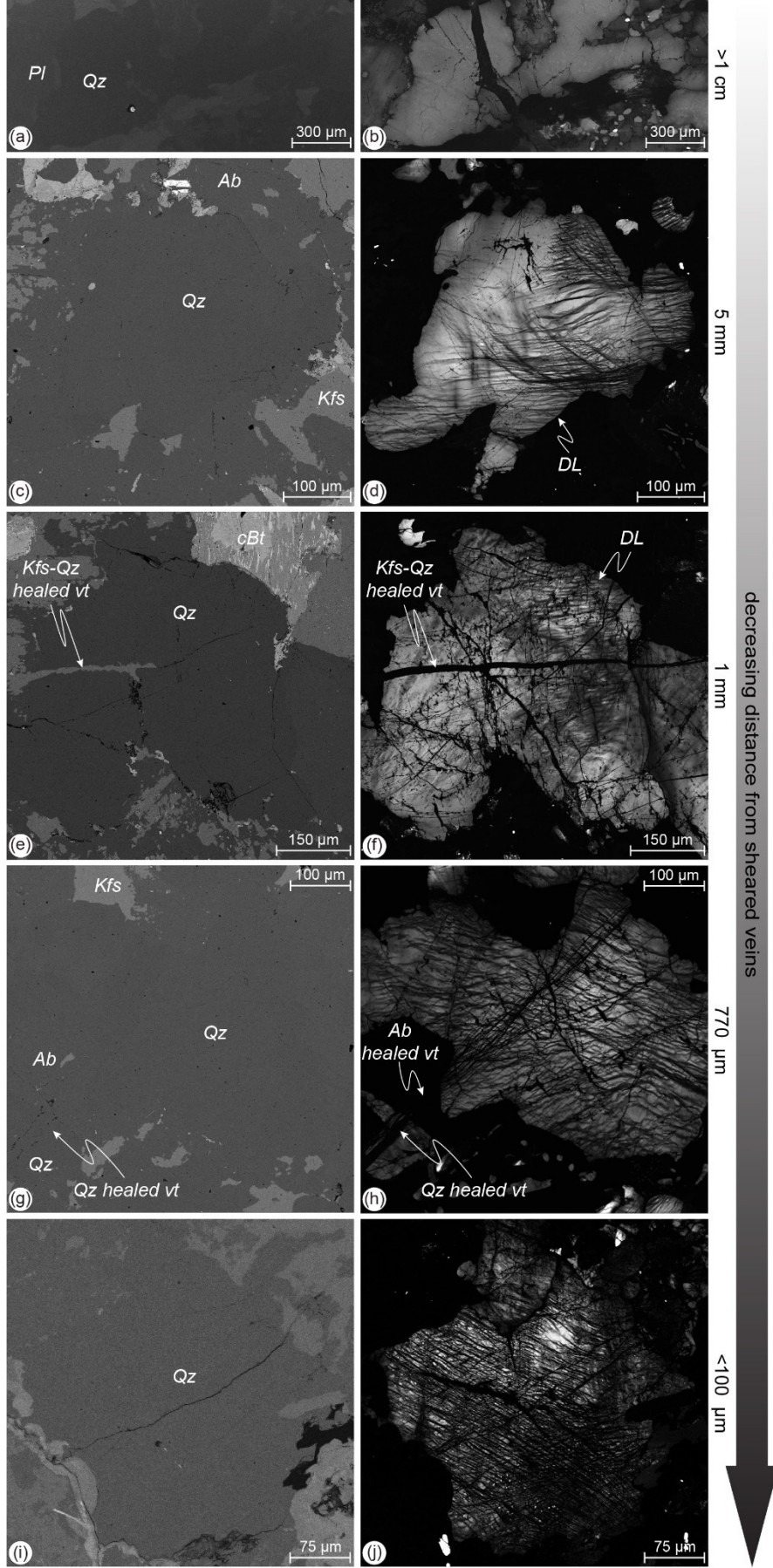

**Figure 4. Quartz microstructures in the weakly-deformed granodiorite (a-b) and in the micro-damage zone of the fault-veins (c-j). BSE images (left column) and their corresponding CL images (right column) with their distance to the vein boundary. Samples 19-37 and 19-**

195 **38. Mineral abbreviations:** *Ab* = albite, *cBt* = chloritized biotite, *Kfs* = K-feldspar, *Pl* = plagioclase, *Qz* = quartz. **(a)** Quartz grains outside the micro-damage zone. **(b)** Undeformed quartz grains show a homogeneous, bright CL signal. **(c, e, g, i)** Quartz grains appear almost undeformed in BSE images. **(d, f, h, j)** Deformed magmatic quartz shows bright to medium, CL grey-shaded domains, which are pervasively cut by interlaced darker deformation lamellae (*DL*). The quartz deformation lamellae are systematically cut by CL-dark veinlets (*healed vt*). Veinlets are healed by quartz ± K-feldspar across quartz grains (see veinlets labelled in e-f) and K-feldspar + albite across feldspar grains (see veinlet warm labelled in g-h), respectively. **(i-j)** Quartz grain close to the vein boundary in the footwall side. 200 In the CL image in (j), the quartz grain appears strongly brecciated (almost pulverized) and is healed by CL-dark quartz.

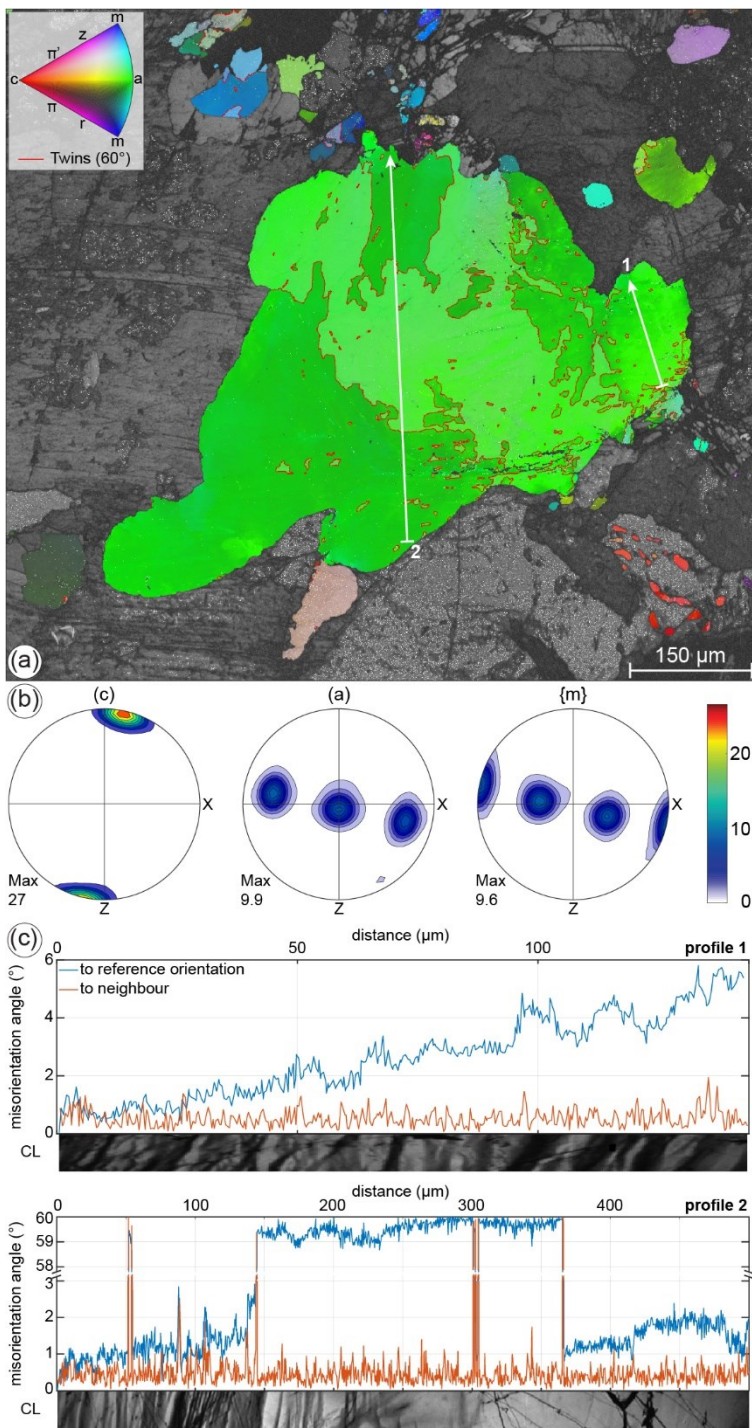

**Figure 5. EBSD analysis of a deformed magmatic quartz in the micro-damage zone. (a)** Inverse Pole Figure (IPF) map, color coded according to IPF legend (Y direction). The analyzed large magmatic quartz grain is the same shown in Fig. 4c-d. The IPF map is overlaid 205 to the orientation contrast image. White lines mark the profiles plotted in (c). **(b)** Contoured pole figures. **(c)** Misorientation profiles. The corresponding CL banding is reported on the bottom of each profile. In profile 2, the Y axis is not in scale.

**4.2 Epidote-rich fault-veins**

The epidote-rich fault-veins have a heterogeneous microstructure (Figs. 3a-c, 3f, 6-7). Samples SQ04-18 and 19-48, which include both sides of the wall-rock surrounding the fault-vein, consist of both undeformed and cataclastic vein domains (Figs. 3c, 3f, 6a).

The undeformed domains consist of idiomorphic, zoned epidote (dark: Al-rich; light: Fe-rich; Table S1) ± prehnite (dark: Al-rich; light: Fe-rich; Table S1), interstitial chlorite ± calcite ± quartz ± K-feldspar, and wall-rock fragments (Fig. 3c, 3f). Undeformed domains are generally present at the outer part of the vein, while the cataclastic domain is at the core (Figs. 3c, 6a). The core of the vein consists of a fine-grained (< 20 µm in size) epidote (ultra)cataclasites including fragments of earlier vein fillings and of the wall-rock (Figs. 2e, 3c).

In samples 19-33, 19-37, 19-38, 19-39, 19-40, 19-43 and 19-46, which only include the footwall wall-rock, the fault-veins consist of layered (proto)cataclasites to ultracataclasites in sharp contact with the topping undeformed vein (Figs. 3a-b, 6a, 7a). Close to the wall-rock, the (proto)cataclasites consist of a fine-grained (< 20 µm in size) matrix of epidote ± prehnite with interstitial chlorite (Fig. 6c-d), including fragments (up to cm in size) of earlier prehnite-epidote veins and wall-rock (Fig. 6a, 6c-d), and some are foliated (Fig. 6e). The ultracataclasites consist of a porous, fine-grained (≤ 500 nm in size) matrix of epidote and prehnite, with

interstitial chlorite, and fragments (up to 100 µm in size) of idiomorphic epidote and prehnite crystals and wall-rock (Fig. 6d, 6f-g). Above the lineated slickensides, multiple vein generations are present (Fig. 2f, 3a-b, 6a, 6d, 6f). Some veins consist of zoned prehnite crystals elongated orthogonal to the vein boundaries (Fig. 6f). Other veins consist of zoned epidote-prehnite crystals, which present localized (ultra)cataclasite layers at the vein boundaries, marking further lineated slickensides (Fig. 6a, 6d). Fragments of magmatic quartz within the fault-veins appear brecciated under CL (Fig. 6h). Micro-fractures are sealed by CL-dark

quartz, which rims the brecciated magmatic quartz fragment (Fig. 6h). This darker rim shows a faint oscillatory zoning in the external part (Fig. 6h). Magmatic quartz included in large (mm in size) wall-rock fragments shows the same deformation features (i.e. deformation lamellae cut by healed veinlets, Figs. 6i-j, S4) as observed in the micro-damage zone (Figs. 3c-e, 4c-j, S1-S2).

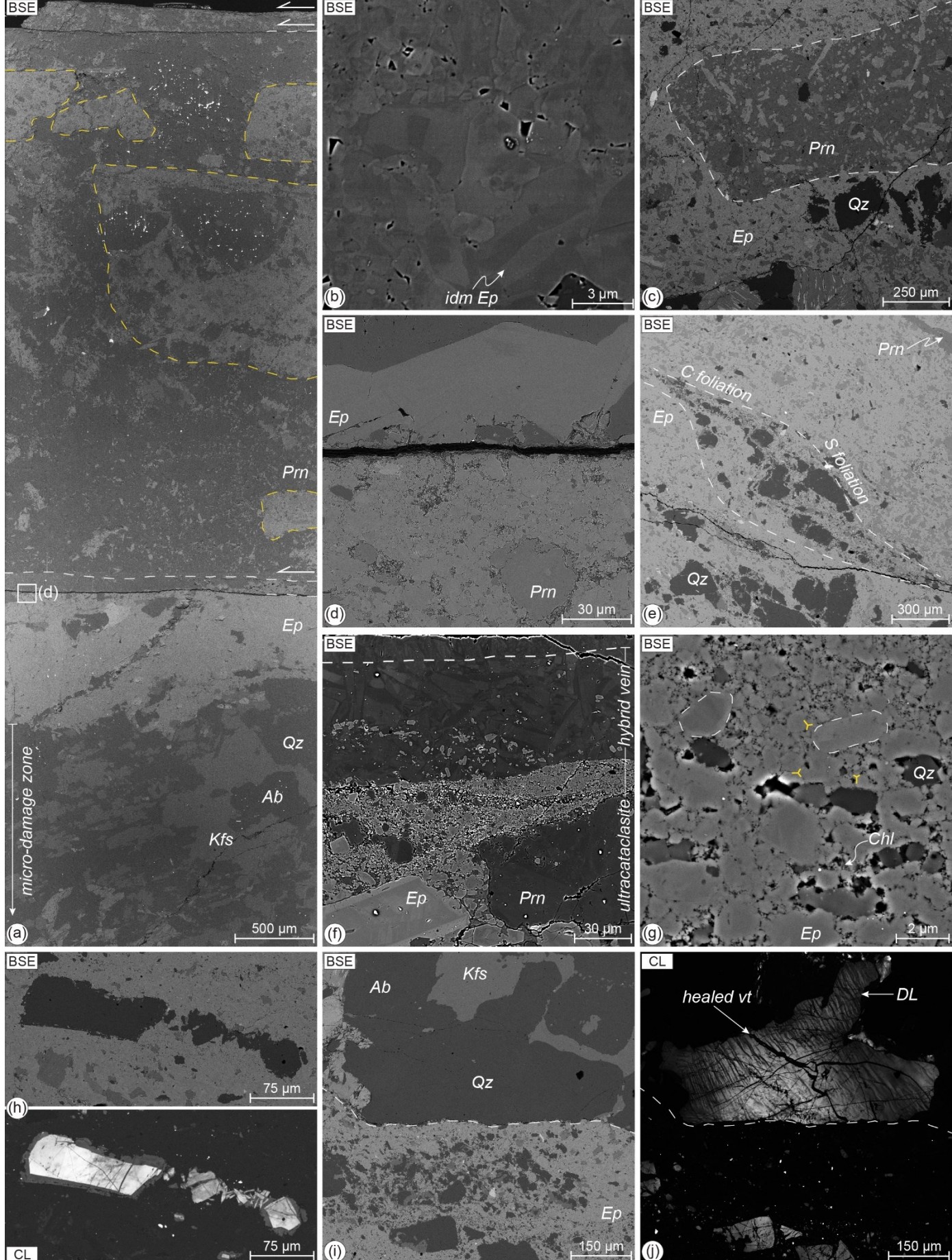

**Figure 6. Microstructures of the epidote-rich fault-veins (samples 19-37, 19-38, 19-46 and 19-48). Mineral abbreviations: *Ab* = albite, *Chl* = chlorite, *Ep* = epidote, *Kfs* = K-feldspar, *Prn* = prehnite, *Qz* = quartz. (a) Overview of an epidote-prehnite fault-vein and associated footwall block. The fault-vein recorded multiple extensional-to-hybrid veining and along vein-boundary cataclasis. The largest vein includes mm-large fragments of earlier veins (dashed yellow lines) within the cataclastic domain. Dashed white lines indicate the top of each vein boundary. The white box indicates the detail shown in (d). (b) Vein filling consisting of idiomorphic zoned epidote (*idm Ep*). (c) Angular fragment of an early prehnite-epidote vein (dashed white line) included in epidote-rich vein protocataclasite. (d) Cataclasite with epidote grains overprinted by an extensional vein with epidote-prehnite crystals. (e) Foliated cataclasite. The sigmoidal clast (dashed lines) consists of wall-rock fragments with elongated tails of finer fragments and epidote grains. (f) Ultracataclasite, defining the slip zone of a discrete polished surface, includes angular fragments of zoned epidote (light grey) and prehnite (dark grey). Multiple events of extensional-to-hybrid veining reactivate the fault-vein. The latter vein is sealed by elongated prehnite crystals (above the white dashed line) and reactivating a hybrid extensional-shear one. (g) Matrix of ultracataclasite consisting of epidote nanoparticles (≤ 500 µm in size). Fragmented idiomorphic crystals of epidote and prehnite (some marked by dashed white lines) are included in the matrix. The ultrafine epidote grains have triple junctions (some highlighted by yellow lines) and pores (<< 1 µm in size), locally filled with chlorite. (h) Quartz fragments within an epidote cataclasite. The quartz fragments are brecciated and rimmed by CL-darker quartz. (i-j) Quartz grains in wall-rock fragments (the larger is marked by the dashed white line) show the same deformation features, i.e. deformation lamellae (*DL*) and healed veinlets (*healed vt*), observed in the micro-damage zone of the fault-veins, shown in Figs. 4c-j, 5, S1-S3. The corresponding OM images are shown in Fig. S4.**

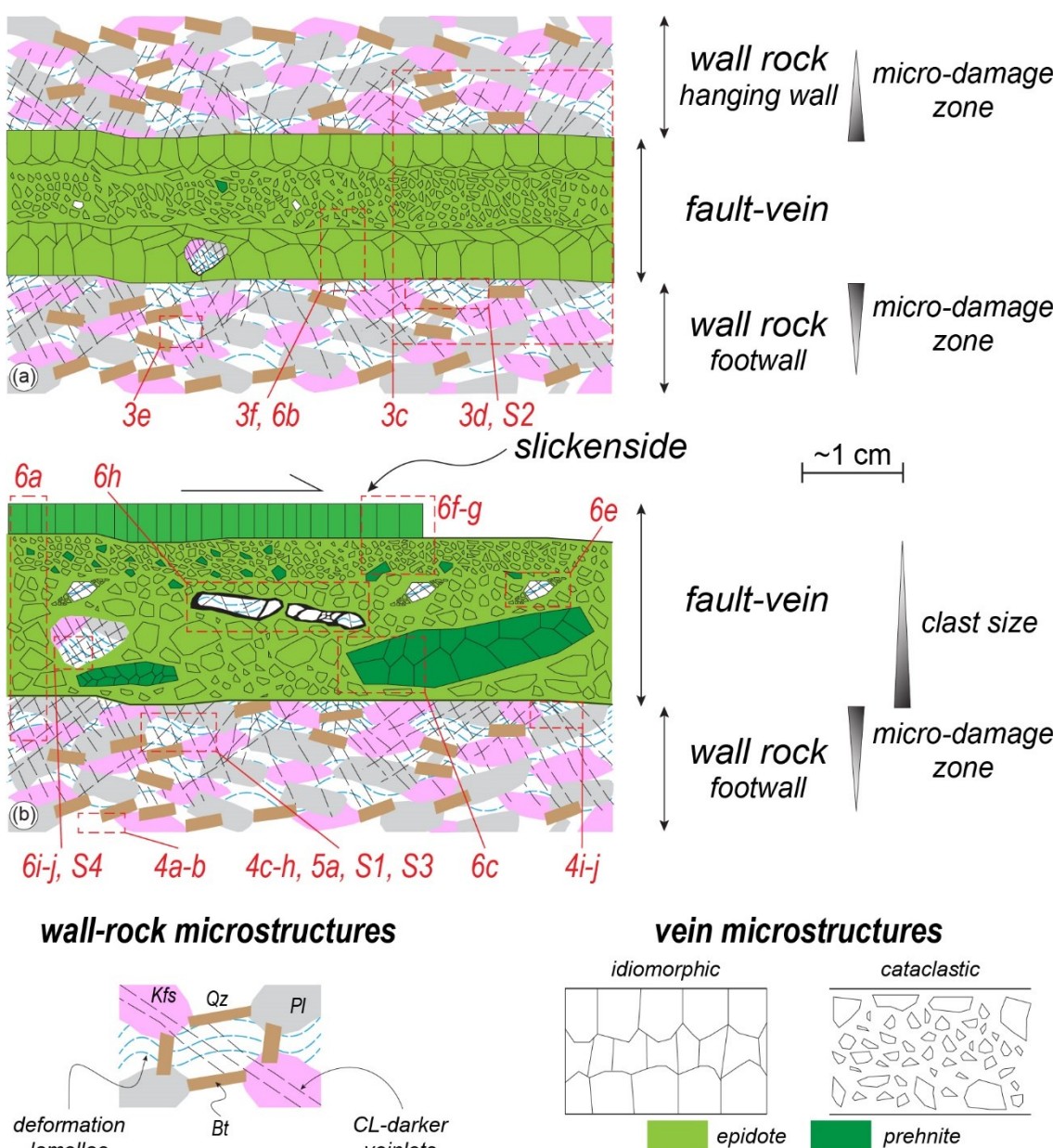

**Figure 7. Schematic illustration summarizing the different microstructures observed in the epidote-rich hybrid extensional-shear veins and associated wall-rock. (a) Fault-veins with both footwall and hanging wall blocks preserved. (b) Fault-veins with only the footwall block preserved.**

**5 Discussion**

The epidote-rich fault-vein networks of the BFZ formed at temperatures ≤ 300 °C (Herrera et al., 2005; Masoch et al., 2022; Olivares et al., 2010), i.e. at conditions close to the brittle-ductile transition for quartz-rich crustal rocks and corresponding to the base of the seismogenic upper crust (Scholz, 2019). Ancient (125-118 Ma) seismicity along the BFZ is attested by pseudotachylytes, produced in a fluid-rich environment (Gomila et al., 2021) along the main segments of the fault system (Masoch et al., 2022, 2021). The epidote-rich fault-vein networks represent a subsidiary linkage set of structures that accommodated slip deficit along, and/or slip transfer between, the main seismogenic segments, during fault system growth (Cembrano et al., 2005; Herrera et al., 2005; Masoch et al., 2022, 2021).

The SEM images document a polyphase deformation history associated with vein array formation, including (i) an initial stage (well-preserved in the wall-rocks nearby the epidote-rich fault-veins, i.e. micro-damage zone) of fracture propagation with local fluid redistribution along micro-cracks, and (ii) following pulses of hydrothermal fluid infiltration, with of epidote ± prehnite, alternating with vein-parallel cataclastic shearing, which shaped the mature architecture of the fault-fracture system. Below, we discuss the microstructural observations and propose a conceptual model for the nucleation (section 5.1) and development (section 5.2) of a highly interconnected fault-fracture network in a seismically-active hydrothermal system (Fig. 8), distinguishing two deformation environments (*rock-buffered* vs. *fluid-buffered*) based on the mineralogy of vein fillings. Lastly, we compare our findings with observations of currently active systems (section 5.3).

**5.1 Wall-rock damage and local fluid redistribution during dynamic crack propagation**

Quartz deformation lamellae and healed veinlets in the micro-damage zone (Figs. 3c-e, 4c-j, 5, S1-S3) of the epidote-rich fault-veins formed at an early stage of development of the hydrothermal fault-vein system (Fig. 8a), as attested by the presence of these microstructures within clasts inside the fault-veins (Figs. 6h-j, S4). Quartz deformation lamellae have been reported in shock-impact rocks (e.g., Carter, 1965) and in exhumed middle-crustal shear zones from the Sesia-Lanzo Zone (Western Alps), associated with other high-stress deformation microstructures (e.g., twinning of jadeite, shattering of garnet), as evidence of upper-crustal seismic ruptures that transiently propagated in the underlying ductile crust (Trepmann and Stöckhert, 2003). A similar interpretation has been proposed for the development of quartz deformation lamellae during the vein emplacement along middle-crustal detachments in the Cyclades belts (Styra-Ochi Unit, Evia Island, Greece) (Nüchter and Stöckhert, 2007, 2008). Deformation lamellae develop in metals deformed at high-strain rates and low temperatures (Drury, 1993). Similarly, they were produced experimentally in natural quartz deformed under high stresses and relatively low temperatures (Trepmann and Stöckhert, 2013). Specifically, in their *kick-and-creep* deformation experiments, Trepmann and Stöckhert (2013) produced basal deformation lamellae with high dislocations density and small misorientation angle (< 2°) at their kick stage (i.e. 400 °C and $10^{-4}$ $s^{-1}$ strain rate), which represented co-seismic loading. In addition, (sub-)basal deformation lamellae in quartz, as those documented in this work (Figs. 5, S1), have been reported to be generated at high differential stresses in the range of 170-420 MPa (Blenkinsop and Drury, 1988; Drury, 1993; Drury and Humphreys, 1988) and in planes of high shear stress (Carter, 1965). On the other hand, quartz deformation lamellae can also develop during comparatively slow tectonic deformation (Derez et al., 2015 and references therein) at greenschist conditions, therefore the conditions at which the studied epidote-rich fault-vein network formed. Notably, the quartz deformation lamellae are genetically associated with the epidote-rich fault-vein emplacement. Indeed, they fade away in the wall-rock (Fig. 4). Consequently, it is unlikely that the deformation lamellae we observed were only produced by long-term slow plastic deformation. As a result, we interpret this quartz deformation microstructure as evidence of transient conditions of high stresses.

During a seismic rupture propagation, a dynamic transient high-stress field is produced in the immediate surrounding of the rupture tip and leads to instantaneous rock failure and pulverization (Faulkner et al., 2011; Okubo et al., 2019; Reches and Dewers, 2005; Vermilye and Scholz, 1998) as recorded in the wall-rock of several exhumed pseudotachylyte-bearing faults (e.g., Di Toro et al., 2005; Mancktelow et al., 2022; Petley-Ragan et al., 2019; Toffol et al., 2024). In contrast to seismic ruptures propagating at velocities of $1-4 \times 10^3$ m/s, micro-cracks may also propagate at extremely low velocities (sub-seismic: $10^{-9}-10^{-4}$ m/s) by sub-critical crack growth driven by stress corrosion (Atkinson and Meredith, 1987). Sub-critical crack propagation is particularly efficient in silicate-built rocks in the presence of pressurized water, which maintains crack connectivity, and at high fluid temperatures (T ≥ 200°C), therefore at the ambient conditions during formation of the fault-vein networks described in this study. However, sub-critical crack propagation cannot explain the high-stress perturbations recorded by the quartz deformation lamellae in the wall-rock surrounding the epidote-rich fault-veins (Trepmann and Stöckhert, 2013) (Figs. 3c-e, 4c-j, 5, S1-S3). Thus, in the relatively small-displacement (< 1.5 m) and up to 10s-m-long faults and hybrid fractures of the epidote-rich fault-vein networks, we interpret the occurrence of deformation lamellae in the wall-rock quartz to reflect the transient high-stress pulse associated with rupture tip propagation at seismic speeds during initial fracturing (Fig. 8a). Blenkinsop and Drury (1988) proposed a similar interpretation for the formation of this low-temperature intra-crystalline deformation microstructure found in the damage zone of the Bayas Fault hosted in quartzites (Cantabrian Zone, Variscan Orogen, Spain). In addition, independent observations from the same study area of the geometry of epidote-rich fault-veins and their haloes, which taper towards the tips (Faulkner et al., 2011), suggest that at least some of these fault-veins record dynamic rupturing, consistently with our findings. Moreover, the high dislocation density of quartz deformation lamellae could have facilitated local fluid infiltration (Christie et al., 1964; Drury, 1993) and Ti resetting at lower concentrations (Bestmann et al., 2021) as indicated by the darkening of CL signal along the deformation lamellae (Figs. 4, S1-S3).

The veinlets sharply and systematically crosscut the quartz deformation lamellae (Figs. 3-4, S1-S2), increase in spatial density towards the vein boundary (Fig. 4), are mostly oriented at high angle with respect to the vein boundary (Figs. 3d, 4h, 4j, S2), and are healed by the minerals (quartz and K-feldspar, K-feldspar and albite across quartz and feldspar grains, respectively) of the crosscut wall-rock (Figs. 4c-j, 5, S1-S2). Moreover, at the vein boundary in the footwall wall-rocks, the deformed magmatic quartz is strongly brecciated (Fig. 4i-j), resembling *in-situ* shattered or pulverized fault rocks found in exhumed upper to mid-lower crustal seismic fault zones (e.g., Fondriest et al., 2015; Johnson et al., 2021; Mancktelow et al., 2022; Mitchell et al., 2011; Ostermeijer et al., 2022). We therefore infer that the healed veinlets also resulted from wall-rock damage associated with the dynamic stress field during seismic rupture tip propagation. Micro-fracturing and rapid healing of seismic faults has been documented in pseudotachylyte-bearing faults hosted in quartzo-feldspathic rocks and referred to the initial stage of seismic rupture propagation (Bestmann et al., 2016, 2012; Mancktelow et al., 2022). Williams and Fagereng (2022) reviewed the role of quartz precipitation in healing seismic faults during the seismic cycle at different environmental conditions and by different mechanisms (e.g., fluid advection, fluid depressurization, dissolution-precipitation creep, frictional heating). The authors observed that, at crustal conditions similar at which the epidote-rich fault-vein networks formed (i.e. temperature ≤ 300 °C and 3-7 km depth), micrometer-thick veins can be completely healed by quartz in a timeframe spanning from days to hundreds of years, depending on the mechanisms involved in quartz precipitation. The quartz-healed veinlets are hundreds of µm in length (Figs. 3c-e, 4d, 4f, 4h, 4j, S1d, S2) and up to 15 µm in thickness (Fig. 4f, 4h) with most veinlets ~2-3-µm-thick (Figs. 4d, 4f, 4h, 4j, S1d, S2). The co-seismic opening of these micro-cracks induced a sudden decrease of pore-fluid pressure ranging from near-lithostatic to sub-MPa levels (e.g., Brantut, 2020; Cox, 2016; Sibson, 1992a, 1992b) that likely resulted in quartz (super)saturation, and eventually into local fluid vaporization (Amagai et al., 2019; Williams, 2019), and in rapid precipitation of amorphous silica (Amagai et al., 2019). Assuming the healing rates estimated by Williams and Fagereng (2022) (see their Fig. 8 and their discussion), the  veinlets could

have reasonably healed in a timeframe as long as tens of years (considering the largest veinlets), during the co- to post-seismic phase. Moreover, the veinlet filling is controlled in composition by the crosscut wall-rock minerals (quartz-K-feldspar-albite; Figs. 3d-e, 4c-j, S1-S2), discarding any extensive fluid advection from external reservoirs (Williams and Fagereng, 2022). This observation also indicates that the co-to-post-seismic micro-fracture formation and healing occurred in a *rock-buffered* system, where percolation of external hydrothermal fluids or fluid redistribution was still minor, owing to the still immature stage of development of a fully interconnected network of permeable fractures and more conspicuous fluid circulation (Fig. 8a). In summary, the microstructures preserved in the deformed magmatic quartz in the proximity of epidote-rich fault-veins resulted from dynamic propagation of small seismic ruptures and co- to post-seismic healing of a newly-produced micro-fracture network. Both low-temperature crystal-plasticity (quartz deformation lamellae) and micro-fracturing accommodated the high-stress conditions around a propagating seismic rupture (Fig. 8a).

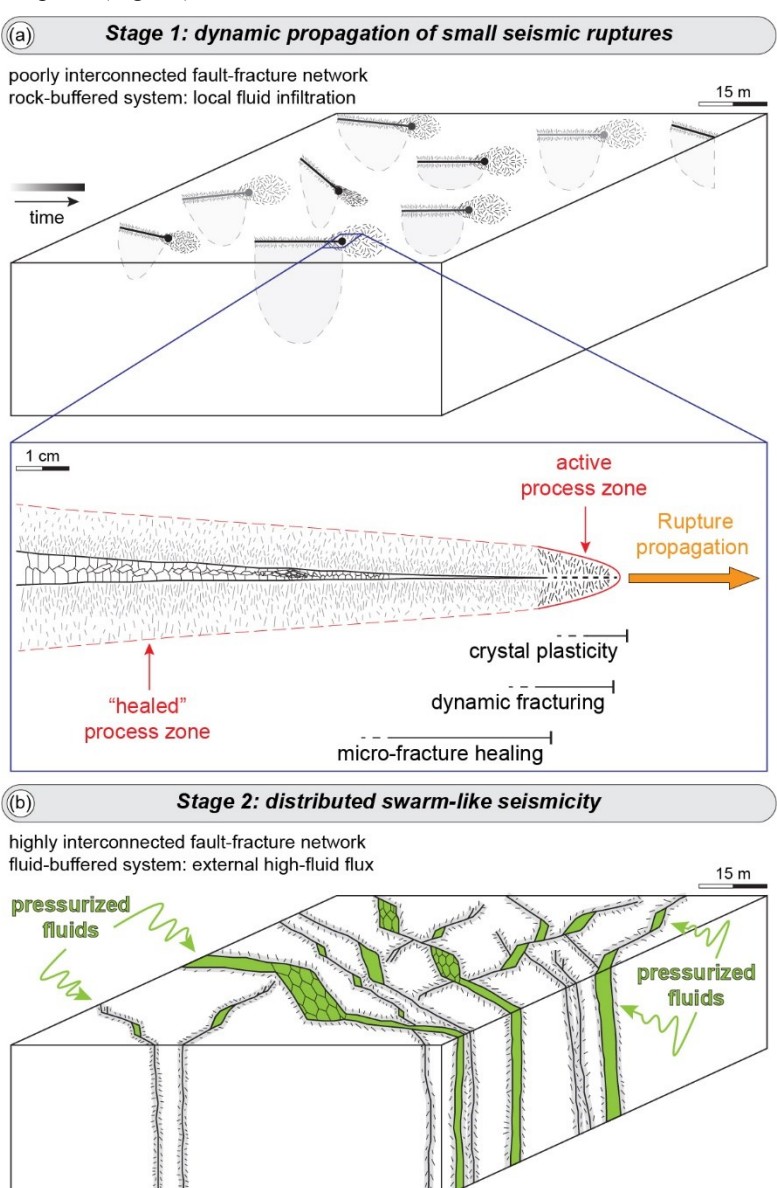

**Figure 8. Conceptual model summarizing the development of the seismically-active hydrothermal system recorded in the studied epidote-rich fault-vein networks. (a) Stage 1: initial stages of dynamic propagation of small seismic ruptures. The fault-fracture network is poorly interconnected, and, in turn, fluid circulation is relatively low and at cm-scale (*rock-buffered system*). The blue box marks the zoom at the crack tip and shows the sequences of deformation processes that recorded the initial stages, well preserved in the wall-rocks, of seismic rupture propagation. (b) Stage 2: distributed swarm-like seismicity (*fluid-buffered system*). Highly-interconnected fault-fracture**

**networks allow the ingression of overpressured fluids leading to multiple swarm-like earthquake sequences, well recorded in the fault-veins. The cyclic deformation sequence is driven by fluid pressure fluctuations as illustrated in Fig. 9.**

**5.2 Pore pressure oscillations in a highly connected hydrothermal (fluid-buffered) fault-fracture network**

The epidote-rich veining and shearing postdate the initial short-term co- to post-seismic deformation recorded in the deformed wall-rock magmatic quartz, as discussed in the previous section. The initial fracturing and associated wall-rock damage was

precursory to development of a more robust external fluid ingression within the initially low-permeability crystalline rocks (Fig. 8). Robust fluid ingression was accompanied by a switch from the initially fluid-poor *rock-buffered* system to a *fluid-buffered* one (Masoch, 2023) (Fig. 8b).

The epidote-rich fault-vein networks show cyclic and mutually overprinting events of extensional veining and shearing (Figs. 2d, 3a-c, 6d, 6f). Cataclasites include fragments of earlier veins (Figs. 2d, 3a, 3c, 3f, 6a, 6c, 6f-g), indicating that extensional veining

preceded either hybrid extensional-shear fracturing (Figs. 2d, 3c) or shearing (Figs. 2c, 3a). Cataclasites are overprinted by extensional(-shear) veins, which show cataclastic shearing along vein boundaries (Figs. 3b, 6a, 6d, 6f). Some cataclasites are foliated (Fig. 6e) suggesting that slip likely occurred by aseismic fault creep (e.g., Angiboust et al., 2015; Chester and Chester, 1998; Rutter et al., 1986). On the other hand, most cataclasites display suspended clasts of wall-rocks and earlier veins (Figs. 2d, 3a, 6a, 6c, 6h-j) similar to the microstructures observed in fluidized cataclasites and breccias, which have been interpreted as

markers of co-seismic slip (e.g., Berger and Herwegh, 2019; Cox and Munroe, 2016; Fondriest et al., 2012; Masoch et al., 2019; Muñoz-Montecinos et al., 2021; Smith et al., 2008). Moreover, some epidote-rich fault-veins include fragments of earlier prehnite-rich veins (Fig. 6c) and are overprinted by prehnite-rich extensional veins (Fig. 6f); differently, some prehnite-rich fault-veins include fragments of earlier epidote-rich veins (Fig. 6a). Thus, the mineral assemblage and microstructures of the fault-veins indicate that polyphase deformation history recorded by the fault-veins was coeval.

The overprinting between extensional veining and shearing can be interpreted with the use of $\lambda - \Delta\sigma$ failure mode diagrams (Cox, 2010), where $\lambda$ is the pore fluid factor ($\lambda = \frac{p}{\sigma_v}$; where $p$ and $\sigma_v$ is the pore fluid pressure and the vertical stress, respectively) and $\Delta\sigma$ is the differential stress ($\Delta\sigma = \sigma_1 - \sigma_3$; where $\sigma_1$ and $\sigma_3$ are the maximum and minimum principal compressive stresses, respectively). At low differential stresses ($\Delta\sigma < 4T$; where $T$ is the tensile strength of the material) and larger rate of increase in pore fluid pressure respect to the increase in tectonic loading, hydraulic fracturing (and extensional veining) occurs before shear

failure (Murrel-Griffith failure criteria; Price and Cosgrove, 1990) (step A, Fig. 9). Opening of extensional fractures prevents further increase in fluid pressure and pressurizes the fracture network. The progressive increase in tectonic-related differential stress leads to hybrid extensional-shear failure (step B, Fig. 9) up to shear failure (step C, Fig. 9), causing stress drop and fault depressurization (step D, Fig. 9). The progressive increase in tectonic-related differential stress could be achieved because the NE-, SW- and NW-dipping small-displacement epidote-rich vein arrays are (near-)optimally oriented with respect to the tectonic

stress field (i.e. nearly subvertical-oriented compression direction; Cembrano et al., 2005; Veloso et al., 2015). The described deformation cycle can repeatedly occur if the system is dominated by an increase in the rate of fluid pressure larger than an increase in the rate of tectonic loading (Cox, 2016; Phillips, 1972). Moreover, as in the studied samples, extensional veining precedes either hybrid and shear failure, it is likely that epidote and prehnite sealing promotes recovery of cohesive strength in the timescales of rupture recurrence during a swarm (Fig. 9). However, we cannot rule out that part of the cyclic deformation history recorded by

the epidote-rich fault-veins is the result of deformation events unrelated to the coupled evolution of fluid pressure and tectonic differential stress.

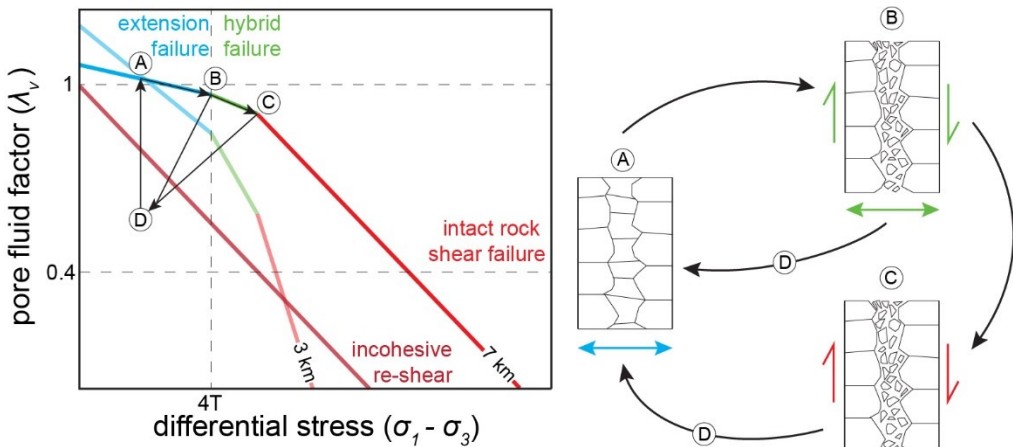

**Figure 9. λ – Δσ diagram (left) and cartoon (right) illustrating the deformation cycle (steps A to D) governing seismicity during the swarm stage. Failure curves represented for the minimum and maximum formation depths of the epidote-rich fault-vein network. The schematic λ – Δσ diagram illustrates the fluid pressure vs. tectonic stress paths recorded by the fault-veins, which show cyclic fluid-driven extensional-to-hybrid veining and shearing. The evolution of fluid pressure and stress states along with recovery of cohesive strength due to epidote and prehnite precipitation control the temporal evolution and deformation path of swarm sequence till fluid depletion.**

### 5.3 Implications for natural fluid-driven earthquake swarms

Earthquake swarms are characterized by a spatiotemporal clustering of large number of small magnitude events, without a clear triggering mainshock (Mogi, 1963). Such a behavior requires external mechanisms driving seismicity, among which fluid diffusion and aseismic slip are the preferred ones (e.g., De Barros et al., 2020; Lohman and McGuire, 2007; Vidale and Shearer, 2006). Recent studies revealed that both processes can coexist with fluid diffusion favoring the occurrence of aseismic slip, which triggers seismicity by stress transfer ahead of the slip front (e.g., Danré et al., 2022b; Guglielmi et al., 2015). The occurrence of swarms is also controlled by the complexity of fault systems, such as fault linkages, step-overs, or fluid-rich fracture zones (e.g., Essing and Poli, 2022; Legrand et al., 2011; Poli et al., 2017; Ross et al., 2020, 2017; Shelly et al., 2022). For instance, thanks to high-precision earthquake relocation, Shelly et al. (2022) documented that two conjugate sets of strike-slip faults well-oriented with respect to the far-field stress were activated during the swarm-like 2020 Maacama sequence. Most earthquakes had moment magnitude $M_W < 1$ and localized in overstepping segments of the Maacana Fault (Northern California). Moreover, swarm-like sequences produce both non-double-couple (i.e. isotropic) and double-couple events in the same period of time, resulting from co-seismic fault opening (dilation) and shearing, respectively (e.g., Legrand et al., 2011; Shelly et al., 2013).

Geological observations at fault system scale (Figs. 1-2; Masoch et al., 2022) and microscale (Figs. 3-6, S1-S3) show several analogies with the characteristics of earthquake swarms. At Stage 1 (Fig. 8a), we infer the early development of a fault-fracture mesh within a low-strained and low-permeability rock volume, producing the pathways for the ingression of external pressurized hydrothermal fluids sustaining the swarmogenic activity of Stage 2 (Figs. 8b-9). The microstructures found in the micro-damage zones of the fault-veins (i.e. quartz deformation lamellae and quartz-healed veinlets; Figs. 3-5, S1-S2) are consistent with the propagation of small-in-magnitude earthquakes (i.e. dynamic ruptures), possibly also accompanied by quasi-static crack growth (Stage 1, Fig. 8a). At Stage 2 (Fig. 8b), the fault-fracture network progressively became hydraulically more interconnected (Fig. 8b). Cyclic fluid pressure fluctuations drove widespread epidote precipitation and development of the epidote-rich fault-vein mesh (Figs. 8b-9). It is necessary to bear in mind that our microstructural observations represent different snapshots of the polyphase spatio-temporal development of the BFZ, which accommodated a minimum displacement of ~1 km through several episodes of seismic and aseismic slip (Cembrano et al., 2005; Masoch et al., 2022). Consequently, the deformation sequences we described could have occurred multiple times during the BFZ lifetimes, leading to the formation of the extensive hydrothermal fault-vein networks associated with the crustal Coloso Duplex. Thus, the studied epidote-rich fault-vein arrays possibly recorded multiple

swarm-like sequences (Stage 2). Meanwhile, each sequence could either (i) be associated with a preliminary stage producing the same wall-rock deformation described in section 5.1, at least, during early stages of the fault system development, or (ii) reactivate earlier epidote-rich fault-vein arrays.

We associate the Stage 2 with the activation of a swarmogenic system (Fig. 8b) as suggested by the following analogies between our geological observations and earthquake swarms:

1. *Fault geometric complexity*: the small-displacement (< 1.5 m) fault-veins are located at geometric complexities, such as fault linkages and intersections (Fig. 1b), within the crustal Coloso Duplex (Cembrano et al., 2005; Masoch et al., 2022) (Fig. 1a). The fault-vein system is arranged into sets (i.e. NW-, NE- and SW-dipping fault-veins; Fig. 1c) (near-)optimally oriented with respect to the local-stress field (i.e. subvertical-oriented $\sigma_1$; Cembrano et al., 2005; Veloso et al., 2015). Many works have shown that fault geometric complexities are the *loci* for the development of earthquake swarms (e.g., Legrand et al., 2011; Ross et al., 2020, 2017), commonly activing fault-fracture networks well-oriented with the stress field (Shelly et al., 2022). Moreover, this structural arrangement forms a honey mesh-like fault network at the scale up to 100s of meter (Figs. 1b, 2a-b), which is the fault-fracture geometry commonly inferred to be activated during swarms (Hill, 1977; Sibson, 1996).

2. *Fluid diffusion within the fault system*: faulting was driven by incoming pressurized fluids within the fault system (section 5.2) and the fault-veins recorded cyclic extensional-to-hybrid veining and shearing (Fig. 2, 3a-c, 3f, 6a-g). They might be interpreted as non-double-couple (crack opening) and double-couple (shear fracture) fracture processes occurring in swarm-like sequences (e.g., Legrand et al., 2011; Shelly et al., 2013). Bursts of short-lasting (tens to thousands of seconds) fluid pressure variations trigger repeated small earthquakes along active fault systems (Collettini, 2002; Essing and Poli, 2022; Piana Agostinetti et al., 2017). Similarly, such a repeated condition of fluid (over-)pressurization in short timespans drives the deformation cycle (i.e. crack opening followed by along vein-boundary slip) recorded in the veins (Figs. 3a-c, 6a, 6c-g) and described by the diagram in Fig. 9.

3. *Coexistence of both aseismic and seismic slip*: the fault-veins accommodated either aseismic fault slip, as attested by foliated cataclastic horizons (Fig. 6e), and possible seismic fault slip, as documented by the occurrence of suspended clasts within cataclasites (Figs. 2d, 6a, 6c, 6h-j), mutually overprinting crack opening (i.e. extensional veins) (Fig. 6a, 6f). The occurrence of both slip behaviors, coupled with fluid pressure diffusion, has been recently observed in the both natural swarm-like sequences (Danré et al., 2022b) and fluid-injection experiments (Guglielmi et al., 2015).

4. *Small scale length*: the small-displacement fault-veins extend up to tens of meters in length (Figs. 1b, 2a-b) and have a thickness up to 2-3 cm (Figs. 2, 3c), resulted from multiple events of crack opening and fracture shearing (Figs. 2, 3a-c, 6a, 6c-d, 6f-g). Considering that each crack opening episode results in dilatant slip ranging from tens to hundreds of $\mu$m (Fig. 6a, 6f), these are equivalent to micro-seismic events with $-2 < M_W < 0$ (Wells and Coppersmith, 1994), which is the magnitude range typical of earthquake swarms (Mogi, 1963).

## 6 Conclusions

The extensive epidote-rich fault-vein networks of the damage zone of Bolfin Fault Zone and of the Coloso Duplex, at larger scale, are exceptionally well-exposed over tens of square kilometers in the Atacama Desert (Northern Chile) (Fig. 1). The fault-vein networks are spatially distributed around major transtensional pseudotachylyte-bearing faults of the duplex, and consist of fault-veins with lineated slickenside, extensional and hybrid veins, and dilatant breccias (Fig. 2). Based on microstructural analysis, we document that the wall-rocks in proximity to small-displacement (< 1.5 m) fault-veins initially experienced dynamic high stresses

related to the propagation of small seismic ruptures in a poorly connected fault-fracture system with limited fluid infiltration (Figs. 3c-e, 4-5, 8a, S1-S2). Instead, the epidote-rich fault-veins recorded cyclic crack opening and either seismic or aseismic shearing dominated by fluid pressure fluctuations in a mature and highly interconnected fault-fracture system (Figs. 2, 3a-c, 3f, 6, 8b, 9).

As a consequence, the epidote-rich fault-vein networks of the Bolfin Fault Zone and, at larger scale, of the Coloso Duplex represent the mature architecture of a fault-fracture system in a high-fluid flux hydrothermal setting. Thus, the Coloso Duplex is interpreted as a fossil example of an upper-crustal seismogenic hydrothermal system, which could potentially have generated multiple fluid-driven earthquake swarms.

**Acknowledgements**

This research was funded by ERC CoG project NOFEAR 614705 to GDT. SM acknowledges Fondazione CARIPARO (PhD scholarship), Fondazione Ing. Aldo Gini and the School of Science of Università degli Studi di Padova (staying in Chile). GP and GDT acknowledge funding from PRIN 2020WPMFE9. MF received funding from the EU Horizon 2020 MSCA-IF DAMAGE (No. 839880), from NextGenerationEU (REACT project) and from the 2021 STARS Grants@Unipd programme (STIFF project). RG have received funding from the European Union's Horizon 2020 research and innovation program under the Marie
Skłodowska-Curie grant agreement No 896346 (FRICTION). JC acknowledges support by Fondecyt grant #1210591, on fluid transport through vein networks and at fault intersections in the crust. We thank Leonardo Tauro and Silvia Cattò (thin sections), Nicola Michelon (scans of thin sections), Stefano Castelli (scans of polished samples), and Jacopo Nava (technical assistance using electron microscopy). SM thanks Giovanni Toffol for help with MTEX and the fruitful discussions. SM acknowledges Stephen Cox and Christie Rowe for fruitful comments regarding his PhD thesis. We acknowledge Catriona Menzies, Randy Williams and
Giancarlo Molli for commenting an early version of this manuscript. We thank the Editors Federico Rossetti and Florian Fusseis for the editorial work, and two anonymous reviewers for their constructive and fruitful comments.

**Data availability**

The EBSD data are available in the repository published by Masoch and Pennacchioni (2023).

**Author contribution**

**Simone Masoch**: Conceptualization, Formal analysis, Investigation, Writing – Original Draft, Visualization, Funding acquisition. **Giorgio Pennacchioni**: Conceptualization, Investigation, Writing – Review & Editing, Supervision. **Michele Fondriest**: Conceptualization, Writing – Review & Editing. **Rodrigo Gomila**: Writing – Review & Editing. **Piero Poli**: Conceptualization, Writing – Review & Editing. **José Cembrano**: Writing – Review & Editing, Supervision, Funding acquisition. **Giulio Di Toro**: Conceptualization, Writing – Review & Editing, Supervision, Project administration, Funding acquisition.

**Competing interests**

The authors declare that they have no conflict of interest.

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
