# Peer review of "Earthquake swarms frozen in an exhumed hydrothermal system (Bolfin Fault Zone, Chile)"

_EGUsphere, 2024_

## Author Comment (AC1)

We thank the reviewer for their comments which we will address in the revised version of our manuscript and which will help us to improve its quality. See the attached file for our reply to the **reviewer's comments (bold font)**.

**Reviewer 1**

**The manuscript "Earthquake swarms frozen in an exhumed hydrothermal system (Bolfin Fault Zone, Chile)" submitted by Masoch et al. describes microstructures from epidote-prehnite sealed hybrid veins within granitoids from the Bolfin Fault Zone, Chile. Magmatic quartz clasts in the veins contain various deformation microstructures: (1) deformation lamellae characterized by an undulating, lamellar change in extinction position in transmitted polarized light and darker CL, (2) healed microcracks characterized by fluid inclusion trails in single quartz grains and dark CL as well as (3) fractures in quartz sealed with feldspar and quartz (veinlets).**

These deformation features are firstly observed in the host rock. The deformation lamellae and the quartz-healed veinlets (i.e. thin veins) are present within the magmatic quartz nearby (i.e., within 1 cm distance) the epidote-rich veins. The veinlets are sealed by epitaxially quartz ± feldspar.

**These deformation microstructures occur in association with shear zones related to the Bolfin Fault Zone. Together with the observation that veins contain fragments of former veins, the quartz deformation microstructures are taken to record cyclic cataclasis and sealing and is interpreted as an ancient seismogenic hydrothermal system. The topic is timely and of significant interest to the readers of Solid Earth.**

**However, from my point of view, the presentation, interpretation and discussion of the deformation lamellae, healed microcracks and veinlets in magmatic quartz, where especially the deformation lamellae are taken as one important argument for transient high stresses, should be improved before publication. The specific comments are as follows:**

**A better presentation and correlation/distinction between *deformation lamellae, healed microcracks and veinlets* in the magmatic quartz would be necessary, as the microstructures record different processes. Especially, a correlation of the lamellar change in extinction position (i.e. "deformation lamellae") in polarized light micrographs with the marked change in CL shown in the impressive Figures 4 d, f, h, would be important. The lamellar change in extinction position is shown from four quartz grains in the images Fig. 3d, e, S1a, b and S2, which are of relatively low magnification. The correlation with the CL images is, however, only shown for the quartz grains in Figs. S1a, b, d and S2/Fig. 6j, where the deformation lamellae are not that well visible, as the thin sections are 100μm thick.**

The slight difference in orientation across the deformation lamellae is hardly seen in the optical images especially in 100 μm-thick thin sections. The deformation lamellae are instead clearly highlighted by CL imaging (as shown in the "impressive Figures 4 d, f, h" to use the reviewer's words) and the crystallographic mismatch typical of deformation lamellae is well captured by EBSD mapping (Fig. 5). To account for the reviewer request of a better documentation of the deformation lamellae, we therefore added the EBSD maps of the other examples of deformation lamellae in the Supplementary Material. These images are much more useful than the optical images.

**Furthermore, Fig. 3d shows fluid inclusion trails in a quartz grain, thus interpreted as healed microcracks. It would be good to also show a CL image of this grain, to see the difference in CL related to the healed microcracks and to the deformation lamellae.**

We will add the corresponding CL image of the microstructure in Fig. 3d as Supplemental Material.

**As both, healed microcracks and deformation lamellae, appear to be characterized by darker CL in relation to the host, a distinction from CL images alone is not possible.**

In CL images, the healed microcracks and the deformation lamellae show quite distinct features: (i) the deformation lamellae resent a gradual dark shade, while the healed microcracks are homogeneously black; the deformational lamellae have a blurred boundary in contrast with the neat, sharp boundary of the healed microcracks; (ii) the healed micro-cracks continue in the neighbor minerals as quartz and feldspars; (iii) there is also a clear overprinting relationship between deformation lamellae and healed micro-cracks: indeed, the latter systematically crosscut the deformation lamellae. We will add this list of distinctive microstructural features in the text.

**The BSE images Fig. 4 a, c, g, e, are not helpful, as they show no contrast. Maybe BSE images at higher magnification and better contrast would help to show at least the "veinlets" sealed also with feldspar, as described in the text (e.g. line 155-160) with reference to the Figures 4f, 6i, 6j, S1, which however, are not really helpful to distinguish the veinlets from the healed microcracks, as no other phases are indicated.**

The meaning of the BSE images was exactly to highlight the complete healing of veinlets (i.e., healed micro-cracks) within quartz by new quartz. We agree that probably the four BSE images are too many and not of great help and we will add some figures with details of the quartz –feldspar gain boundary to show the extension of the healed veinlets in quartz and feldspars as Qz-Felds healed micro-cracks.

**Maybe some confusion arises also because the text does not distinguish between epitaxially healed microcracks in quartz, e.g. characterized by "fluid inclusion trails" in a quartz grain with one main extinction position as shown in Fig. 3d, and "quartz-healed" veinlets with other phases like albite and K-Fsp (e.g. line 155-160). It would be good to present also data of the other phases in the veinlets, I did not find a figure showing these?**

We will rephrase the main text to make this difference clear between the healed veinlets (in quartz) and the Qz-Feld-healed micro-cracks (extending from quartz into the neighbor feldspars.

**The *EBSD data* presented in Figure 5 are not suited to characterize the slight oscillatory change in crystallographic orientation, which would be expected across the deformation lamellae. The misorientation profiles are not very specific and could also represent the usual "noise" of deformed quartz with internal misorientation/undulatory extinction. The gradual increase in misorientation angle to the reference orientation in profile 1 (blue line) is probably not related to the deformation lamellae. A correlation of the misorientation profiles with GROD, GOS or relative misorientation maps would be necessary to relate the slight changes in orientation across the lamellae.**

The misorientation across deformation lamellae is usually very small (1-3°). Thus, the misorientation is affected by the precision of EBSD measurements. However, it is quite clear the systematic variation in

crystallographic orientation which overlaps with the CL dark banding. To better highlight this match, we will add a band reporting the CL variations across the profile at the base of the misorientation profile of Fig 5c.

**Were more EBSD measurements performed then those from Fig. 5? Are the observed deformation lamellae generally of (sub)basal orientation and what is the angle to the basal plane? This information would be important for the discussion.**

Many more EBSD maps have been produced. If considered necessary, we can add other maps in the supplementary material. Only basal deformation lamellae have been identified in the studied samples.

**The discussion on the deformation lamellae in *Chapter 5.1* is misleading: The two planar shock effects in quartz, Planar Fracturs (PFs) and Planar deformation features (PDFs), from meteorite impactites are very different from deformation lamellae. PFs are cleavage cracks in quartz and are typically not associated with a change in extinction position, as deformation lamellae are. PDFs are very straight, following specific crystallographic planes (they can be curved but only if the grain orientation is changing respectively), very fine lamellar (typically without a change in extinction position, i.e. without misorientation along the lamellae) and occur in sets of different orientations. Basal PDFS in quartz are mechanical Brazil twins (only resolvable by TEM) and rhombohedral PDFs are characterized by localized transformations from/to diaplectic glass. A comparison to the undulating dark lamellae in CL observed here appears very distracting and misleading from my point of view.**

The discussion of PDFs was actually added as response to a former friendly review and was not included in the original draft. We agree that this discussion is misleading and will delete it.

**Instead, a much more specific and careful discussion of (sub)basal deformation lamellae, short wave length undulatory extinction and fine extinction bands in quartz, the indication of the respective formation processes (relevance of dislocation glide, pile up of dislocations, influence of microcracking (!) etc.) as well as a discussion on the deformation conditions (transient high stresses?) would be very important and relevant here.**

We will add a more extended discussion of the (sub)basal deformation lamellae, short wave length undulatory extinction and fine extinction bands in quartz.

**Further comments:**

**Line 30-32: I think that also Nüchter and Stöckhert, 2007, 2008 (references see below) are very relevant studies on the topic of transient permeability and fluid flow related to seismic activity recorded by quartz veins, which should be discussed here.**

We agree and will add this reference, and related discussion, in the main text.

**Line 78, please rephrase, as this might be mistakeable. I assume that here the initiation of the fault system is meant and not the nucleation of an earthquake.**

We will rephrase the sentence to make our idea clearer.

**Line 135, Fig. 2a) the hybrid extensional-shear veins and alteration halos are not well visible in the figure**

We will label them to make their more visible.

**Line 153-170 please rewrite this paragraph and describe the microstructures more specifically: e.g., I recommend to not use "planar features" which is very unspecific and maybe mistakeable by the very specific term "planar deformation features" in shocked quartz from meteorite impactites. This is especially as the microstructures shown in Fig. 3e and 4 are not really planar except maybe of the healed microcracks in Fig. 3d, see specific comment 1 above. Usually the term "deformation lamellae" is referring to lamellar undulatory extinction with the lamellae subparallel to the basal plane, i.e. the plane at high angle to , as described here. The term short wave length undulatory extinction is referring to wavy, lamellar undulatory extinction also subparallel to other crystallographic planes (m, r, z).**

As explained in the response above, we agree that the PDFs should not be discussed in our manuscript.

**Line 155-160: I strongly recommend to distinguish fractures in quartz sealed also with other phases (i.e. veins, or if you wish veinlets) from epitaxially healed microcracks in quartz (Fig. 3d)**

Actually, we believe that this distinction in made quite clearly, but we will further improve the text to avoid any misunderstanding.

**line 165: are all observed deformation lamellae of (sub)basal orientation? What is the angle of the lamellae to the basal plane?**

See previous response.

**Line 169-170: the epitaxy of the healed microcracks is better to see in Fig. 3d, by the same extinction position of the grain (but please label the healed microcracks, see below). Where is the veinlet in Fig. 5a, which is corresponding to the grain in Fig. 4d, please indicate the veinlet in both images. In the BSE image also no veinlet is visible, which should indicate the presence of different phases, as feldspar. Again, an optical micrograph could be much more helpful…**

As commented above, we will substitute some of the BSE images of Fig. 4 with some detailed BSE images showing micro-cracks healed by quartz and feldspars.

**3 d, e: Please indicate also the healed microcracks in Fig. 3d. It would be very interesting to see the CL images of these two grains, especially Fig. 3d, to see the difference in CL related to the healed microcracks and to the deformation lamellae.**

We will add some arrows to indicate the deformation lamellae and the healed micro-cracks (i.e., veinlets). We will also add the corresponding CL images in the Supplementary Material.

**4: The undulating lamellae dark in CL are very impressive. Yet, the figure does not allow to distinguish between deformation lamellae, veinlets or healed microcracks, as all three microstructures are obviously characterized by the wavy lamellae of dark CL and in BSE all three microstructures do not show up. I would expect that at least the different phases and grains in veinlets should be visible in BSE, but for that a better contrast/resolution would help. Also, in optical micrographs, all three microstructures should easily be distinguishable. Thus, I strongly recommend to exchange the BSE images with meaningful optical micrographs.**

See response above.

**Line 185 caption to Fig. 4: what do you expect from BSE image? The z-contrast can show other phases, e.g. in your veinlets, but no veinlets filled with albit/K-Fsp are shown? The orientation contrast may show deformation microstructures, but not in this resolution…, see also specific comment 1 above.**

See response above.

**Line 195/EBSD data, see specific comment 2 above.**

See response above.

**Line 216: fig. 5h? fig. 6h?**

Fig. 6h.

**6 b: the idiomorphic zoned epidote does not really show up in the figure?**

It is. We will add some labels to make it clearer.

**6e S-C foliation?**

We will improve the labels showing the S and C planes.

**6g) idiomorphic crystals? Triple junctions, of course there are triple junctions in a 2D picture of a polyphase aggregate, what is the relevance?**

Some broken idiomorphic epidote crystals are marked by the dashed white lines. Triple junctions can be related to pressure solution creep.

**6 i, j where are the veinlets, as referenced in line 219, there are wavy dark lamellae in the CL images, but whether these are deformation lamellae, healed microcracks or veinlets with other phases does not get clear from these two images.**

We will label them to make these figures clearer.

**Lines 260-285 please rewrite this paragraph, see specific comment 3 above**

We agree with the reviewer. See detailed responses above.

**line 397, where are the quartz-healed veinlets in S1?**

They are marked by fluid inclusion trails. We will label them.

**Figure S1, 2: the deformation lamellae are hard to be resolved in the micrographs of the 100µm-thick thin sections.**

See response above about the comparison of the spatial resolution of SEM-CL and optical microscope images.

**Recommended references:**

**Nüchter, J.-A., and B. Stöckhert (2007), Vein quartz microfabrics indicating progressive evolution of fractures into cavities during postseismic creep in the middle crust, J. Struct. Geol., 29, 1445–1462.**

**Nüchter, J.-A., and B. Stöckhert (2008), Coupled stress and pore fluid pressure changes in the middle crust: Vein record of coseismic loading and postseismic stress relaxation, Tectonics, 27, TC1007, doi:10.1029/2007TC002180.**

---

## Author Comment (AC2)

We thank the reviewer for their comments which we will address in the revised version of our manuscript and which will help us to improve its quality. See the attached file for our reply to the **reviewer's comments (bold font)**.

**Reviewer 2**

**The manuscript by Masoch et al deals with the microstructures from a fault zone in northern Chile related with tectono-magmatic and hydrothermal activity. The authors claim that the deformation modes recorded therein could be representative of processes at stake in active swarm systems. Even though the idea of relating the features visible in this fault zone is not totally new and already proposed in the previous papers from the same group on this locality, I found the study well conducted, adequately documented and rather convincing. There are probably several ways of interpreting the finite structures recorded in this fault zone. Future work will tell if those are indeed swarms -or not-. The discussion and the genetic model is mostly supported by the observations even though some re-wording and clarification is needed in places (see below). I also believe that the microstructures should perhaps be described with greater care. Overall I recommend publication after moderate revisions.**

**I think that the authors should better explain the part on the internal versus external fluid infiltration, and locate the epidote-bearing domains in their figure 8. Because epidote formation requires the incoming of large amounts of calcium, I believe that a better discussion of this aspect would bring water to their mill when it comes to discussing fluid infiltration after the first fracturing event. Have similar epidote-rich veins been described elsewhere in Northern Chile (or elsewhere) as a consequence of deeper emplacement/cooling of a plutonic body? Can you discard the possibility that the fluid source comes from the currently downgoing subducting plate?**

By using hydrogen and oxygen stable geochemistry, we constrained that epidote-forming fluids derived from surficial, evaporated (i.e., basin-derived) fluids. Part of our dataset is presented and discussed in Masoch (2023) (see chapter 4 of the thesis). Thus, we discard the hypotheses proposed by the reviewer. However, we investigated different fault-zone rocks (e.g., chlorite-rich cataclasite, pseudotachylytes, epidote-rich fault-veins, hydrothermal breccias) of the Bolfin Fault Zone and hydrogen-bearing minerals of the magmatic rocks cut by the fault. Consequently, we rather prefer to not enclose our isotopic data to this manuscript because they constitute an independent study about the conditions of fluid-rock interaction during fault zone growth.

**The interpretation of fault zone structures in terms of slip velocity is hazardous. It is currently impossible to tell whether breccias or cataclasites result from slip at seismic strain rates in the absence of pseudotachylytes. I suggest a more careful writing.**

We agree with reviewer. However, ancient seismicity along the BFZ is documented by widespread occurrence of pseudotachylytes and, as documented in this work, microstructures typical of high-stress conditions (i.e., wall-rock quartz deformation lamellae and wall-rock pulverization) associated with the described sheared fault-veins.

**Detailed points:**

**L.14: fault-veins / epidote faulting: what do you mean? Please clarify. Not clear enough for an abstract.**

With fault-veins, we mean hybrid-shear veins. We will make these aspects clearer in the abstract.

**L.17-21: the abstract lacks material that enables understanding how you came to these conclusions.**

We will rephrase and include material to the abstract to make it more clear.

**L.65: linkage zone? What do you mean?**

We mean "zone of fault linkage". We will modify it in the next version of the manuscript.

**L.129: replace by 'EBSD data was processed using the MTEX…'**

We will modify the text as suggested by the reviewer.

**L.132: why such a low (6 nA) current?**

Because these working conditions were the best to get detailed resolute images.

**L.135: you mean alkali devolatilization?  (instead of migration)**

Yes, we will modify the text in the revised version of our manuscript.

**L.202: (Al-rich; light: Fe-rich) there is something missing here**

(*Dark*: Al-rich; light: Fe-rich).

**L.211: show the pores!**

Pores are shown in Fig. 6d, 6f and 6g.

**L.216: there is no figure 5h! check again the figure calls. Do you mean 6h?**

Yes. We thank the review for noting this typo.

**L.267: 'high stresses': how high?**

Trepmann and Stöckhert (2013) reached confined pressure up to 2.7 GPa in their experiments.

**L.269: how slow?**

We mean strain rate slower than what expected for co-seismic deformation.

**L.282: 'the latter hypothesis': clarify to which hypothesis you are referring to**

We refer to the hypothesis that most quartz deformation lamellae we observed are possibly related to high-stress conditions rather than slow low-temperature crystal plasticity. We will make this point clearer.

**L.295-298: It is not clear to me how high stresses can be reached here in hybrid veins, since mode ii veins by definition require higher fluid pressures than pure shear veins**

Indeed, we refer to the high-stress perturbation at the crack tip propagating at seismic speeds. We will clarify this in the revised version.

**L.305: rephrase: too long and not clear**

We will rephrase the sentence to make it clearer.

**L.314-317: Yes, but this study deals with dissolution precipitation of the host metasediments towards the host. In this study you suggest permeability increments associated with fluid advection.**

We agree with the reviewer but a previous reviewer suggested to take in consideration these works and we discuss them.

**L.326: any evidence for pressure-solution at this stage that could account for elemental redistribution?**

No, we did not observe any evidence of pressure-solution in the wall-rocks.

**L.351: see Angiboust et al. (2015, G-cubed) for another example where an epidote-rich cataclazed fault system is described, forming as a consequence of transients increase in pore fluid pressure in a sheared system. See also Oncken et al. (2021, geosphere) for further evidence of foliated cataclasites as a key fault zone material in deep plate boundary systems. See also Muñoz-Montecinos et al. (EPSL, 2021).**

We will take in consideration these papers in the revised version of our manuscript.

**L.353: foliated, fluidized cataclasites and breccias may also form at sub-seismic strain rates (see Oncken et al., 2021). I would be more careful in the writing here.**

We will be more careful in our statement.

**L.370: check syntax in this sentence (perhaps you mean 'by an increase of the rate of fluid pressure…')**

Yes, thanks for noting this misleading sentence. We will modify the text in the revised version of our manuscript.

**L.372: which type of deformation events?**

Independent events of extensional faulting followed by fault reactivation.

**L.384: coexist (no 's')**

Yes, thanks for noting this typo. We will modify the text in the revised version of our manuscript.

**L.430: the presence of suspended clasts within cataclasite should not be viewed alone as a solid evidence for seismic slip**

We agree with the reviewer. Indeed, we specified that this microstructure is a possible marker of paleo-earthquake.

**Bird & Spieler (2004, rev. Min. Geochem) could be a useful reference when it comes to demonstrating that your veins were once part of a supra-plutonic environment.**

See response above about fluid sources.

**Figure 1: how do you define the 'weakly fractured unit'? is there a statistical criterion or is it purely arbitrary? Also, explain better your difference between chloritized/Fractured and chlorite-rich cataclastic: foliated and massive, in the caption of the map.**

The structural units forming the BFZ are presented in detail in Masoch et al. (2022) and their classification was based on a quantitative analysis. In that work, we defined the different units based on the following structural features: preservation of original magmatic features of the host rocks, alteration degree of the host rocks, spacing of fractures, veins and faults, relatively abundance of veins and faults, mineral assemblage sealing and decorating veins and faults, and clast/matrix proportion in the fault rocks.

**Figure 3: better label minerals and features as explained in the caption**

We will improve the labelling.

**Figure 4: what are the analytical conditions used for obtaining these CL images?**

The working conditions at which the CL images were acquired are described in the methods section of our manuscript.

**Figure 5b: how many points considered for this pole figure? figure 5a: why not showing the misorientation map instead? It would be more useful here. What is the Y direction and what is the reference frame?**

We plotted 833 points in the pole figure. Fig. 5a shows the orientation of the deformation lamellae compared to the crystallographic orientation of the host quartz grain. Microstructural observations were conducted parallel to the fault lineation (X direction) and orthogonal to the fault/vein wall (X-Y plane), as specified in the methods section.

**Figure 8: part (a) title: propagation (not progation). The three lines (crystal plasticity, dynamic fracturing …) are not well positioned and not very easy to understand in the frame of this sketch.**

We will make the figure clearer. We thank the reviewer for highlighting the typo.

**Figure 9: ok but this figure does not consider the reactivation of the fault**

We will modify the figure taking in consideration fault reactivation.

**References**

Masoch, S., 2023. Structure, evolution and deformation mechanisms of crustal-scale seismogenic faults (Bolfin Fault Zone, Northern Chile). PhD thesis. Università degli Studi di Padova.

Masoch, S., Fondriest, M., Gomila, R., Jensen, E., Mitchell, T.M., Cembrano, J., Pennacchioni, G., Di Toro, G., 2022. Along-strike architectural variability of an exhumed crustal-scale seismogenic fault (Bolfin Fault Zone, Atacama Fault System, Chile). J. Struct. Geol. 165, 104745. https://doi.org/10.1016/j.jsg.2022.104745

Trepmann, C.A., Stöckhert, B., 2013. Short-wavelength undulatory extinction in quartz recording coseismic deformation in the middle crust – an experimental study. Solid Earth 4, 263–276. https://doi.org/10.5194/se-4-263-2013

---

## Author Response (AR1)

Padova, 31 October 2024

Dear Editor,

Thank you for editorial work regarding our manuscript eguspere-2024-1841; we also thank the two reviewers for their constructive comments. Please find below **their comments in bold font** and our response to all their comments in normal font.

We hope that with this rebuttal letter we addressed the points raised by the two reviewers and the revised version of the manuscript will be considered suitable for publication in your journal.

With our best regards,

Simone Masoch[*], Giorgio Pennacchioni, Michele Fondriest, Rodrigo Gomila, Thomas Mitchell, Piero Poli, José Cembrano, Giulio Di Toro

*Dipartimento di Geoscienze,*
*Università degli Studi di Padova,*
*Via Giovanni Gradenigo 6,*
*35131 Padova - ITALY*
*Mail: simone.masoch@unipd.it*

**Reviewer 1**

We thank the reviewer 1 for their constructive comments which help us to improve the quality of our manuscript.

**The manuscript "Earthquake swarms frozen in an exhumed hydrothermal system (Bolfin Fault Zone, Chile)" submitted by Masoch et al. describes microstructures from epidote-prehnite sealed hybrid veins within granitoids from the Bolfin Fault Zone, Chile. Magmatic quartz clasts in the veins contain various deformation microstructures: (1) deformation lamellae characterized by an undulating, lamellar change in extinction position in transmitted polarized light and darker CL, (2) healed microcracks characterized by fluid inclusion trails in single quartz grains and dark CL as well as (3) fractures in quartz sealed with feldspar and quartz (veinlets).**

These deformation features are firstly observed in the host rock. The deformation lamellae and the healed veinlets (i.e. thin veins) are present within the magmatic quartz nearby (i.e., within 1 cm distance) the epidote-rich veins. The veinlets are epitaxially healed by quartz $\pm$ K-feldspar across quartz grains and K-feldspar + albite across feldspars grains, respectively. See below for implement to the revised version of the manuscript.

**These deformation microstructures occur in association with shear zones related to the Bolfin Fault Zone. Together with the observation that veins contain fragments of former veins, the quartz deformation microstructures are taken to record cyclic cataclasis and sealing and is interpreted as an ancient seismogenic hydrothermal system. The topic is timely and of significant interest to the readers of Solid Earth.**

**However, from my point of view, the presentation, interpretation and discussion of the deformation lamellae, healed microcracks and veinlets in magmatic quartz, where especially the deformation lamellae are taken as one important argument for transient high stresses, should be improved before publication. The specific comments are as follows:**

**A better presentation and correlation/distinction between *deformation lamellae, healed microcracks and veinlets* in the magmatic quartz would be necessary, as the microstructures record different processes. Especially, a correlation of the lamellar change in extinction position (i.e. "deformation lamellae") in polarized light micrographs with the marked change in CL shown in the impressive Figures 4 d, f, h, would be important. The lamellar change in extinction position is shown from four quartz grains in the images Fig. 3d, e, S1a, b and S2, which are of relatively low magnification. The correlation with the CL images is, however, only shown for the quartz grains in Figs. S1a, b, d and S2/Fig. 6j, where the deformation lamellae are not that well visible, as the thin sections are 100µm thick.**

The slight difference in orientation across the deformation lamellae is hardly seen in the optical images especially in 100 µm-thick thin sections. The deformation lamellae are instead clearly highlighted by CL imaging (as shown in the "impressive Figures 4 d, f, h" to use the reviewer's words) and the crystallographic mismatch typical of deformation lamellae is well captured by EBSD mapping (Figs. 5-S1e-g, S3b-c). To account for the reviewer request of a better documentation of the deformation lamellae, we therefore added EBSD maps of the other examples of deformation lamellae in the Supplementary Material. These images are much more useful than the optical images.

[Figure]

**Figure S1**. Quartz grain within the micro-damage zone of a sheared vein (sample 19-38). Distance from the vein boundary ~770 μm. White arrows = deformation lamellae, yellow arrows = healed veinlets;

orange arrows = trace of misorientation profile in (h). (a) Plane-polarized light micrograph. (b) Cross-polarized light micrograph. Note that the thin section thickness is 100 µm. Thus, interference colors in the cross-polarized light image are not the regular ones observable in 30-µm-thick thin sections. (c) and (d) BSE and corresponding CL images. (e) Inverse Pole Figure (IPF) map, color coded according to IPF legend (Y direction). The IPF map is overlaid to the orientation contrast image. (f) Contoured pole figures. (g) Misorientation profile with corresponding CL banding.

[Figure]

**Figure S3.** Quartz grain within the micro-damage zone of a sheared vein (sample 19-38). Distance from the vein boundary ~5 µmm. (a) CL image. (b) Inverse Pole Figure (IPF) map, color coded according to IPF legend (Y direction). The IPF map is overlaid to the orientation contrast image. The black line marks the profile in (c). (c) Misorientation profile with corresponding CL banding.

**Furthermore, Fig. 3d shows fluid inclusion trails in a quartz grain, thus interpreted as healed microcracks. It would be good to also show a CL image of this grain, to see the difference in CL related to the healed microcracks and to the deformation lamellae.**

We added the corresponding CL image of the microstructure in Fig. 3d as Supplemental Material (new Fig. S2).

[Figure]

**Figure S2.** Corresponding CL image of the quartz grain shown in Fig. 3d. The deformed magmatic quartz (*Qz*) shows bright to medium, CL grey-shaded domains, which are pervasively cut by interlaced darker deformation lamellae (*DL*). The deformed quartz grains neighbor albite grains (*Ab*) are cut by CL-dark veinlets (*veinlets*). The veinlets correspond to the micro-fractures outlined by fluid inclusion trails (yellow arrows in Fig. 3d). Sample SQ04-18.

**As both, healed microcracks and deformation lamellae, appear to be characterized by darker CL in relation to the host, a distinction from CL images alone is not possible.**

In CL images, the healed microcracks and the deformation lamellae show quite distinct features: (i) the deformation lamellae resent a gradual dark shade, while the healed microcracks are homogeneously black; the deformational lamellae have a blurred boundary in contrast with the neat, sharp boundary of the healed microcracks; (ii) the healed micro-cracks continue in the neighbor minerals as quartz and feldspars; (iii) there is also a clear overprinting relationship between deformation lamellae and healed micro-cracks: indeed, the latter systematically crosscut the deformation lamellae. We added this list of distinctive microstructural features in the text.

Old: "*The granodiorite adjacent to epidote-rich fault-veins is turned into reddish alteration haloes, up to 4 cm in thickness (Figs. 2a, 2c-d, 3a-c), associated with (i) replacement of magmatic plagioclase by albite + epidote, and of magmatic biotite and amphibole by chlorite ± opaques (Fig. 3a-c), (ii) pervasive micro-fracturing, filled with epidote ± chlorite ± prehnite (Fig. 3c), and (iii) deformation of the magmatic quartz (Figs. 3c-e, 4, S1). Quartz deformation microstructures include planar features (Fig. 3d-e, S1), up to 10-μm-thick, visible in CL by the darker shade crosscutting the bright to medium grey-shaded host quartz (Fig. 4c-f, S1). The planar features appear straight under the optical microscope (Figs. 3d-e, S1) and in the CL images (Figs. 4d, S1) and become interlaced and wavy when approaching the vein boundaries (Fig. 4f). These quartz deformation microstructures are crosscut by thin (up to 15-μm-thick) micro-fractures healed by quartz ± K-feldspar ± albite (hereafter referred as "quartz-healed" veinlets),*

*across quartz and K-feldspar grains (Fig. 4c-f). These veinlets are outlined by fluid inclusion trails under the optical microscope (Fig. 3c-e), show a homogeneous dark CL shade (Fig. 4d, 4f, 4h) and are oriented at high angle with respect to the vein boundary (Figs. 3d, 4f). These deformation microstructures (hereafter referred to as "micro-damage zone") fade away from the fault-veins and disappear at distances ≥ 1 cm (Fig. 4a-b). In the micro-damage zone, the quartz-healed veinlets increase in spatial density towards the fault-veins (Figs. 3c-e, 4c-f), while no apparent change in density of interlaced planar features is observed. In the footwall block, at < 100 µm distance from the sharp vein boundary, the magmatic quartz is strongly brecciated and healed by CL-dark grey-shaded quartz (also surrounded by epitaxial rim of CL-dark quartz; Fig. 4g-h).*

*EBSD maps of the quartz show that the planar features visible in CL are oriented nearly orthogonal to the <c> axis (Fig. 5a-b) and correspond to a minor crystallographic misorientation (< 2-3°; see profiles in Fig. 5c) with respect to the host grain. These features are typical of deformation lamellae (Fairbairn, 1941; Trepmann and Stöckhert, 2003), either referred to as short-wavelength undulatory extension (Trepmann and Stöckhert, 2013) or fine extinction bands (Derez et al., 2015). Therefore, quartz planar features will be referred to hereafter as deformation lamellae. The EBSD maps also show that and the quartz-healed veinlets overgrew in epitaxial continuity with the host magmatic quartz (Fig. 5a)."*

New: *"The granodiorite adjacent to epidote-rich fault-veins is turned into reddish alteration haloes, up to 4 cm in thickness (Figs. 2a, 2c-d, 3a-c), associated with (i) replacement of magmatic plagioclase by albite + epidote, and of magmatic biotite and amphibole by chlorite ± opaques (Fig. 3a-c), (ii) pervasive micro-fracturing, filled with epidote ± chlorite ± prehnite (Fig. 3b-c), and (iii) deformation of the magmatic quartz (Figs. 3c-e, 4-5, S1-S3). Quartz deformation microstructures include (1) deformation lamellae (see EBSD data description below) (Fig. 3d-e, S1a-d), up to 10-µm-thick, visible in CL by the darker shade crosscutting the bright to medium grey-shaded host quartz (Fig. 4c-h, S1d, S2, S3a). The deformation lamellae appear straight under the optical microscope (Figs. 3d-e, S1a-b) and in the CL images (Figs. 4d, S1d, S2-S3a) and become interlaced and wavy when approaching the vein boundaries (Fig. 4h). The quartz deformation lamellae are systematically crosscut by (2) thin (up to 15-µm-thick) micro-fractures (hereafter referred as healed veinlets or veinlets) healed by quartz ± K-feldspar, across quartz grains (Fig. 4e-f), and K-feldspar ± albite when extending across neighbor feldspar grains (Fig. 4i-j). These veinlets are outlined by fluid inclusion trails, across magmatic quartz grains, under the optical microscope (Figs. 3c-e, S1), show a homogeneous dark (i.e. black) CL shade across host quartz and feldspar grains (Fig. 4d, 4f, 4h, 4j, S1d, S2-S3a) and are oriented at high angle with respect to the vein boundary (Figs. 3d, 4h). These deformation microstructures, i.e. quartz deformation lamellae and healed veinlets (hereafter referred to as "micro-damage zone"), fade away from the fault-veins and disappear at distances ≥ 1 cm (Fig. 4a-b). In the micro-damage zone, the healed veinlets increase in spatial density towards the fault-veins (Figs. 3c-e, 4c-j, S1-S2), while no apparent change in density of the quartz deformation lamellae is observed. In the footwall block, at < 100 µm distance from the sharp vein boundary, the magmatic quartz is strongly brecciated and healed by CL-dark grey-shaded quartz (also surrounded by epitaxial rim of CL-dark quartz; Fig. 4i-j).*

*EBSD maps of the quartz show that the deformation lamellae visible in CL are oriented nearly orthogonal to the <c> axis (i.e. sub-parallel to the basal plane; Figs. 5a-b, S1e-f, S3b) and correspond to a minor crystallographic misorientation (< 2-3°; see profiles in Fig. 5c-d, S1g, S3c) with respect to the host grain, which are the typical features of deformation lamellae in quartz (Carter, 1965; Christie et al., 1964; Drury, 1993; Fairbairn, 1941; McLaren et al., 1970; Trepmann and Stöckhert, 2003; White, 1973). The EBSD maps also show that and the healed veinlets overgrew in epitaxial continuity with the host magmatic quartz (Figs. 5a, S1e)."*

**The BSE images Fig. 4 a, c, g, e, are not helpful, as they show no contrast. Maybe BSE images at higher magnification and better contrast would help to show at least the "veinlets" sealed also with feldspar, as described in the text (e.g. line 155-160) with reference to the Figures 4f, 6i, 6j, S1, which however, are not really helpful to distinguish the veinlets from the healed microcracks, as no other phases are indicated.**

The meaning of the BSE images was exactly to highlight the complete healing of veinlets (i.e., healed micro-cracks) within quartz by new quartz. We agree that probably the four BSE images are too many and not of great help, but however they give information about the mineralogy of the deformation features we present. We added two figures with details of the quartz –feldspar grains to show the extension of the healed veinlets in quartz and feldspars as Qz-Felds healed micro-cracks (new Fig. 4e-f). We improved the quality of Fig. 4 by:

- Adding (e) and (f) where veinlets healed by quartz and K-feldspar crosscut a quartz grain with deformation lamellae (*DL*);
- Labelling the veinlets (healed by quartz across quartz grains) extending to neighbor albite grain, where they are healed by albite, in (g) and (h).

[Figure]

**Figure 4.** Quartz microstructures in the weakly-deformed granodiorite (a-b) and in the micro-damage zone of the fault-veins (c-j). BSE images (left column) and their corresponding CL images (right column) with their distance to the vein boundary. Samples 19-37 and 19-38. Mineral abbreviations: Ab = albite, cBt = chloritized biotite, Kfs = K-feldspar, Pl = plagioclase, Qz = quartz. (a) Quartz grains outside the micro-damage zone. (b) Undeformed quartz grains show a homogeneous, bright CL signal. (c, e, g, i) Quartz grains appear almost undeformed in BSE images. (d, f, h, j) Deformed magmatic quartz shows bright to medium, CL grey-shaded domains, which are pervasively cut by interlaced darker deformation lamellae (DL). The quartz deformation lamellae are systematically cut by CL-dark veinlets (healed vt). Veinlets are healed by quartz ± K-feldspar across quartz grains (see veinlets labelled in e-f) and K-feldspar + albite across feldspar grains (see veinlet warm labelled in g-h), respectively. (i-j) Quartz grain close to the vein boundary in the footwall side. In the CL image in (j), the quartz grain appears strongly brecciated (almost pulverized) and is healed by CL-dark quartz.

**Maybe some confusion arises also because the text does not distinguish between epitaxially healed microcracks in quartz, e.g. characterized by "fluid inclusion trails" in a quartz grain with one main extinction position as shown in Fig. 3d, and "quartz-healed" veinlets with other phases like albite and K-Fsp (e.g. line 155-160). It would be good to present also data of the other phases in the veinlets, I did not find a figure showing these?**

We rephrased the main text to make this difference clear between the healed veinlets (in quartz) and the Qz-Feld-healed micro-cracks (i.e. extending from quartz into the neighbor feldspars), and add more informative images in Fig. 4. See reply above (pages 5-6)

**The *EBSD data* presented in Figure 5 are not suited to characterize the slight oscillatory change in crystallographic orientation, which would be expected across the deformation lamellae. The misorientation profiles are not very specific and could also represent the usual "noise" of deformed quartz with internal misorientation/undulatory extinction. The gradual increase in misorientation angle to the reference orientation in profile 1 (blue line) is probably not related to the deformation lamellae. A correlation of the misorientation profiles with GROD, GOS or relative misorientation maps would be necessary to relate the slight changes in orientation across the lamellae.**

The misorientation across deformation lamellae is usually very small (1-3°; see references cited in the manuscript). Thus, the misorientation is affected by the precision of EBSD measurements. However, it is quite clear the systematic variation in crystallographic orientation which overlaps with the CL dark banding. To better highlight this match, we added a band reporting the CL variations across the profile at the base of the misorientation profiles in Fig. 5c.

[Figure]

**Figure 5.** EBSD analysis of a deformed magmatic quartz in the micro-damage zone. (a) Inverse Pole Figure (IPF) map, color coded according to IPF legend (Y direction). The analyzed large magmatic quartz grain is the same shown in Fig. 4c-d. The IPF map is overlaid to the orientation contrast image. White lines mark the profiles plotted in (c). (b) Contoured pole figures. (c) Misorientation profiles. The corresponding CL banding is reported on the bottom of each profile. In profile 2, the Y axis is not in scale.

**Were more EBSD measurements performed then those from Fig. 5? Are the observed deformation lamellae generally of (sub)basal orientation and what is the angle to the basal plane? This information would be important for the discussion.**

Many more EBSD maps have been produced. We added other maps in the supplementary material (new Figs. S1 and S3 (see them at pages 3-4). Only basal deformation lamellae have been identified in the studied samples.

**The discussion on the deformation lamellae in *Chapter 5.1* is misleading: The two planar shock effects in quartz, Planar Fractures (PFs) and Planar deformation features (PDFs), from meteorite impactites are very different from deformation lamellae. PFs are cleavage cracks in quartz and are typically not associated with a change in extinction position, as deformation lamellae are. PDFs are very straight, following specific crystallographic planes (they can be curved but only if the grain orientation is changing respectively), very fine lamellar (typically without a change in extinction position, i.e. without misorientation along the lamellae) and occur in sets of different orientations. Basal PDFS in quartz are mechanical Brazil twins (only resolvable by TEM) and rhombohedral PDFs are characterized by localized transformations from/to diaplectic glass. A comparison to the undulating dark lamellae in CL observed here appears very distracting and misleading from my point of view.**

The discussion of PDFs was actually added as response to a former friendly review and was not included in the original draft. We agree that this discussion is misleading and deleted it. We improved the discussion in the following way:

Old: "*Quartz deformation lamellae and quartz-healed veinlets in the micro-damage zone (Figs. 3c-e, 4c-h, 5, S1) of the epidote-rich fault-veins formed at an early stage of development of the hydrothermal fault-vein system (Fig. 8a), as attested by the presence of these microstructures within clasts inside the fault-veins (Figs. 6e-g, S2). Quartz deformation lamellae have been reported in shock-impact rocks (e.g., Carter, 1965) and in exhumed middle-crustal shear zones from the Sesia-Lanzo Zone (Western Alps), associated with other high-stress deformation microstructures (e.g., twinning of jadeite, shattering of garnet), as evidence of upper-crustal seismic ruptures that transiently propagated in the underlying ductile crust (Trepmann and Stöckhert, 2003). Deformation lamellae were produced experimentally in natural quartz deformed under high stresses and relatively low temperatures (400 °C) (Trepmann and Stöckhert, 2013). Similarly, they develop in metals deformed at high-strain rates and low temperatures (Drury, 1993). On the other hand, quartz deformation lamellae can also develop during comparatively slow tectonic deformation (Derez et al., 2015 and references therein) at greenschist conditions, therefore the conditions at which the studied epidote-rich fault-vein network formed.*
*Boggs et al., (2001) and Hamers and Drury (2011) proposed several criteria to distinguish tectonic- and shock-related planar deformation features in quartz based on optical and electron CL investigations. They proposed that shock-related deformation lamellae are straight, narrow, well-defined in the host grain and appear in multiple sets. Instead, tectonic-related ones are thicker, curved and without any sharp boundary in the host grain. Quartz deformation lamellae observed in our samples are straight (Figs. 3d-e, 4d, 4f, 5, S1), becoming interlaced, wavy and thicker close to the vein boundary (<< 1 mm; Fig. 4f), are relatively narrow (the few larger are up to 10-µm-thick), sharply crosscut the host quartz showing a dark CL shade (Figs. 4, S1) and, in some grains, multiple sets are present (Fig. 4d, 4f). By comparing these observations with the criteria by Boggs et al. (2001) and Hamers and Drury (2011), the deformation lamellae we observed share more features with those diagnostic (i.e., straight, narrow and*

*well-defined in the host grain) of shock-related deformation lamellae. However, although the deformation lamellae become interlaced, wavy and thicker, once approaching the vein boundary (<< 1 mm; Fig. 4) and thus share some diagnostic features with nonshock-related planar features in quartz (Boggs et al., 2001; Hamers and Drury, 2011), it is unlikely that the deformation lamellae we observed were only produced by long-term slow plastic deformation, instead of transient high-stress conditions. The latter hypothesis is also supported by the genetic association between the epidote-rich fault-vein emplacement and the quartz deformation microstructure, which indeed fades away in the wall-rock (Fig. 4). 
[revised manuscript text omitted]

**Instead, a much more specific and careful discussion of (sub)basal deformation lamellae, short wave length undulatory extinction and fine extinction bands in quartz, the indication of the respective formation processes (relevance of dislocation glide, pile up of dislocations, influence of microcracking (!) etc.) as well as a discussion on the deformation conditions (transient high stresses?) would be very important and relevant here.**

In our samples, we observed only sub-basal deformation lamellae in quartz. Consequently, we modified the discussion improving the conditions at which this quart deformation microstructure could be produced as suggested by the reviewer. See reply to previous point.

**Further comments:**

**Line 30-32: I think that also Nüchter and Stöckhert, 2007, 2008 (references see below) are very relevant studies on the topic of transient permeability and fluid flow related to seismic activity recorded by quartz veins, which should be discussed here.**

We agree and added this reference, and related discussion (see pages 12-13), in the main text. See reply above.

**Line 78, please rephrase, as this might be mistakeable. I assume that here the initiation of the fault system is meant and not the nucleation of an earthquake.**

We rephrased the sentence to make our idea clearer.

Old: "*We interpret these microstructures as evidence of ancient swarm-like activity, from the first stages of dynamic crack propagation to the later cyclic crack opening and both seismic or aseismic slip, driven by fluid pressure fluctuations, within a mature and hydraulically connected fault-fracture system. These exposed fault-vein networks represent a unique geological record of the evolution in space and time of upper-crustal swarm-like seismic sources, from the early nucleation stage to the later development of a mature fault system.*"

New: "*We interpret these microstructures as evidence of ancient swarm-like activity, from the incipient stages of dynamic crack propagation to the later cyclic crack opening and shearing, driven by fluid pressure fluctuations, within a mature and hydraulically connected fault-fracture system. These exposed fault-vein networks represent a unique geological record of the evolution in space and time of a potential upper-crustal swarm-like seismic source, from the incipient stages of the propagation of a newly-produced micro-fracture network to the later development of a mature fault system.*"

**Line 135, Fig. 2a) the hybrid extensional-shear veins and alteration halos are not well visible in the figure**

We added some labels them to make their more visible in Figure 2a.

[Figure]

**Figure 2.** The epidote-rich mesh-like fault-vein network of BFZ. Coin, hammer and cover lens for scale. (a) Hybrid extensional-shear veins and veins are surrounded by a red alteration halo in the damaged wall-rock. Lineated slickensides displace an aplitic dyke accommodating up to tens of centimeters of displacement. WGS GPS location: 23.44368°S, 70.487104°W. (b) Honeycomb mesh structure. WGS GPS location: 23.934255°S, 70.465309°W. Modified from Masoch et al. (2022). (c) Discrete extensional fault surface decorated by epidote slickenfibers. WGS84 GPS location: 23.883944°S, 70.486689°W. Modified from Masoch et al. (2022). (d) Epidote-rich fault-vein including angular fragments of earlier veins (dark green). The fault-vein is reactivated by a whitish calcite-palygorskite vein (boundary on the right side), referable to post-Miocene deformation (see Masoch et al., 2021 for details). Sample 19-33. WGS GPS location: 23.99803°S, 70.44051°W. Modified from Masoch et al. (2022).

**Line 153-170 please rewrite this paragraph and describe the microstructures more specifically: e.g., I recommend to not use "planar features" which is very unspecific and maybe mistakeable by the very specific term "planar deformation features" in shocked quartz from meteorite impactites. This is especially as the microstructures shown in Fig. 3e and 4 are not really planar except maybe of the healed microcracks in Fig. 3d, see specific comment 1 above. Usually the term "deformation lamellae" is referring to lamellar undulatory extinction with the lamellae subparallel to the basal plane, i.e. the plane at high angle to , as described here. The term short wave length undulatory**

**extinction is referring to wavy, lamellar undulatory extinction also subparallel to other crystallographic planes (m, r, z).**

As explained in the response above, we agree that the PDFs should not be discussed in our manuscript and we deleted that former discussion. As suggested by the reviewer, we do not use anymore the terms "planar features" to refer to the deformation lamellae in quartz in the results sections. This to avoid any misleading understanding by the reader and to be more careful with the terminology we use in the manuscript (see the modified result section at pages 5-6). We thank the reviewer for raising this relevant observation to improve our manuscript.

**Line 155-160: I strongly recommend to distinguish fractures in quartz sealed also with other phases (i.e. veins, or if you wish veinlets) from epitaxially healed microcracks in quartz (Fig. 3d)**

Actually, we believe that this distinction in made quite clearly, but we improved the text to avoid any misunderstanding. See reply above (pages 5-6).

**line 165: are all observed deformation lamellae of (sub)basal orientation? What is the angle of the lamellae to the basal plane?**

See previous response at pages 5-6.

**Line 169-170: the epitaxy of the healed microcracks is better to see in Fig. 3d, by the same extinction position of the grain (but please label the healed microcracks, see below). Where is the veinlet in Fig. 5a, which is corresponding to the grain in Fig. 4d, please indicate the veinlet in both images. In the BSE image also no veinlet is visible, which should indicate the presence of different phases, as feldspar. Again, an optical micrograph could be much more helpful…**

As commented above, we implemented Fig. 4 with some detailed BSE images showing micro-cracks healed by quartz and feldspars.

**3 d, e: Please indicate also the healed microcracks in Fig. 3d. It would be very interesting to see the CL images of these two grains, especially Fig. 3d, to see the difference in CL related to the healed microcracks and to the deformation lamellae.**

We added some arrows to indicate the deformation lamellae (white arrows) and the healed micro-cracks (i.e. veinlets; yellow arrows) in Fig. 3d-e. We also added the corresponding CL image of 3d in the Supplementary Material (new Fig. S2).

[Figure]

**Figure 3.** Microstructures of the fault-veins and their associated wall-rock under the optical microscope. Mineral abbreviations: *Ab* = albite, *Cal* = calcite, *Chl* = chlorite, *Ep* = epidote. (a) Plane-polarized light scan of thin section of a lineated fault-vein, showing the spatial distribution of the microstructures observed in the micro-damage zone and in the vein (dashed red boxes). Sample 19-38. (b) Plane-polarized light scan of thin section of a fault-vein recording multiple episodes of extensional-to-hybrid veining and along vein-boundary shearing. Sample 19-46. WGS84 GPS location 23.88428°S, 70.48615°W. The dashed rex box marks the zoom shown in Fig. 6a. (c) Cross-polarized light thin section micrograph of an extensional (*vein filling*) to shear (*cataclasite*) vein displaced by an epidote cataclasite. Quartz grains show undulose extinction in the weakly deformed granodiorite; while, they exhibit deformation lamellae and a dense pattern of fluid inclusion trails in the micro-damage zone. Dashed red

boxes marks the zooms shown in (d-f). Sample SQ04-18. WGS84 GPS location 23.883906 °S, 70.486942 °W. (d) Quartz grain with deformation lamellae (white arrows; _DL_) cut by a dense pattern healed micro-fractures (_healed vt_) outlined by fluid inclusion trails (yellow arrows), whose most pervasive set is oriented perpendicular to the vein boundary. Cross-polarized light micrograph. The corresponding CL image is shown in Fig. S2 (e) Quartz grain with straight and narrow deformation lamellae (white arrows). Cross-polarized light micrograph. (f) Idiomorphic epidote and minor calcite crystals in the outer part of the vein. The inner part consists of a fine-grained cataclasite. Plane-polarized light micrograph.

**4: The undulating lamellae dark in CL are very impressive. Yet, the figure does not allow to distinguish between deformation lamellae, veinlets or healed microcracks, as all three microstructures are obviously characterized by the wavy lamellae of dark CL and in BSE all three microstructures do not show up. I would expect that at least the different phases and grains in veinlets should be visible in BSE, but for that a better contrast/resolution would help. Also, in optical micrographs, all three microstructures should easily be distinguishable. Thus, I strongly recommend to exchange the BSE images with meaningful optical micrographs.**

See responses above at pages 2-6.

**Line 185 caption to Fig. 4: what do you expect from BSE image? The z-contrast can show other phases, e.g. in your veinlets, but no veinlets filled with albit/K-Fsp are shown? The orientation contrast may show deformation microstructures, but not in this resolution…, see also specific comment 1 above.**

See responses above at pages 2-6.

**Line 195/EBSD data, see specific comment 2 above.**

See responses above at pages 2-6.

**Line 216: fig. 5h? fig. 6h?**

Fig. 6h. Correction done.

**6 b: the idiomorphic zoned epidote does not really show up in the figure?**

It is. We added some labels to make it clearer. We added also some labels in (e) and (j) as requested. See points below.

[Figure]

**Figure 6.** Microstructures of the epidote-rich fault-veins (samples 19-37, 19-38, 19-46 and 19-48). Mineral abbreviations: *Ab* = albite, *Chl* = chlorite, *Ep* = epidote, *Kfs* = K-feldspar, *Prn* = prehnite, *Qz* = quartz. (a) Overview of an epidote-prehnite fault-vein and associated footwall block. The fault-vein recorded multiple extensional-to-hybrid veining and along vein-boundary cataclasis. The largest vein includes mm-large fragments of earlier veins (dashed yellow lines) within the cataclastic domain. Dashed white lines indicate the top of each vein boundary. The white box indicates the detail shown in (d). (b) Vein filling consisting of idiomorphic zoned epidote (*idm Ep*). (c) Angular fragment of an early prehnite-epidote vein (dashed white line) included in epidote-rich vein protocataclasite. (d) Cataclasite with epidote grains overprinted by an extensional vein with epidote-prehnite crystals. (e) Foliated cataclasite. The sigmoidal clast (dashed lines) consists of wall-rock fragments with elongated tails of finer fragments and epidote grains. (f) Ultracataclasite, defining the slip zone of a discrete polished surface, includes angular fragments of zoned epidote (light grey) and prehnite (dark grey). Multiple events of extensional-to-hybrid veining reactivate the fault-vein. The latter vein is sealed by elongated prehnite crystals (above the white dashed line) and reactivating a hybrid extensional-shear one. (g) Matrix of ultracataclasite consisting of epidote nanoparticles ($\leq$ 500 μm in size). Fragmented idiomorphic crystals of epidote and prehnite (some marked by dashed white lines) are included in the matrix. The ultrafine epidote grains have triple junctions (some highlighted by yellow lines) and pores (<< 1 μm in size), locally filled with chlorite. (h) Quartz fragments within an epidote cataclasite. The quartz fragments are brecciated and rimmed by CL-darker quartz. (i-j) Quartz grains in wall-rock fragments (the larger is marked by the dashed white line) show the same deformation features, i.e. deformation lamellae (*DL*) and healed veinlets (*healed vt*), observed in the micro-damage zone of the fault-veins, shown in Figs. 4c-j, 5, S1-S3. The corresponding OM images are shown in Fig. S4.

**6e S-C foliation?**

We improved the labels showing the S and C planes. See revised Fig. 6 above.

**6g) idiomorphic crystals? Triple junctions, of course there are triple junctions in a 2D picture of a polyphase aggregate, what is the relevance?**

Some broken idiomorphic epidote crystals are marked by the dashed white lines. Triple junctions can be related to pressure solution creep.

**6 i, j where are the veinlets, as referenced in line 219, there are wavy dark lamellae in the CL images, but whether these are deformation lamellae, healed microcracks or veinlets with other phases does not get clear from these two images.**

We labeled them to make these figures clearer. See new figure at pages 19-20.

**Lines 260-285 please rewrite this paragraph, see specific comment 3 above**

We agree with the reviewer. See detailed responses above (pages 11-13).

**line 397, where are the quartz-healed veinlets in S1?**

They are marked by fluid inclusion trails. We labeled them with some arrows. See new Fig. S1 at page 3.

**Figure S1, 2: the deformation lamellae are hard to be resolved in the micrographs of the 100μm-thick thin sections.**

See response above about the comparison of the spatial resolution of SEM-CL and optical microscope images.

**Recommended references:**

**Nüchter, J.-A., and B. Stöckhert (2007), Vein quartz microfabrics indicating progressive evolution of fractures into cavities during postseismic creep in the middle crust, J. Struct. Geol., 29, 1445–1462.**

**Nüchter, J.-A., and B. Stöckhert (2008), Coupled stress and pore fluid pressure changes in the middle crust: Vein record of coseismic loading and postseismic stress relaxation, Tectonics, 27, TC1007, doi:10.1029/2007TC002180.**
* * *
**Reviewer 2**

We thank the reviewer 2 for their constructive comments which help us to improve the quality of our manuscript.

**The manuscript by Masoch et al deals with the microstructures from a fault zone in northern Chile related with tectono-magmatic and hydrothermal activity. The authors claim that the deformation modes recorded therein could be representative of processes at stake in active swarm systems. Even though the idea of relating the features visible in this fault zone is not totally new and already proposed in the previous papers from the same group on this locality, I found the study well conducted, adequately documented and rather convincing. There are probably several ways of interpreting the finite structures recorded in this fault zone. Future work will tell if those are indeed swarms -or not-. The discussion and the genetic model is mostly supported by the observations even though some re-wording and clarification is needed in places (see below). I also believe that the microstructures should perhaps be described with greater care. Overall I recommend publication after moderate revisions.**

**I think that the authors should better explain the part on the internal versus external fluid infiltration, and locate the epidote-bearing domains in their figure 8. Because epidote formation requires the incoming of large amounts of calcium, I believe that a better discussion of this aspect would bring water to their mill when it comes to discussing fluid infiltration after the first fracturing event. Have similar epidote-rich veins been described elsewhere in Northern Chile (or elsewhere) as a consequence of deeper emplacement/cooling of a plutonic body? Can you discard the possibility that the fluid source comes from the currently downgoing subducting plate?**

By using hydrogen and oxygen stable geochemistry, we constrained that epidote-forming fluids derived from surficial, evaporated (i.e., basin-derived) fluids. Part of our dataset is presented and discussed in Masoch (2023) (see chapter 4 of the thesis). Thus, we discard the hypotheses proposed by the reviewer. However, we investigated different fault-zone rocks (e.g., chlorite-rich cataclasite, pseudotachylytes,

epidote-rich fault-veins, hydrothermal breccias) of the Bolfin Fault Zone and hydrogen-bearing minerals of the magmatic rocks cut by the fault. Consequently, we rather prefer to not enclose our isotopic data to this manuscript because they constitute an independent study about the conditions of fluid-rock interaction during fault zone growth.

**The interpretation of fault zone structures in terms of slip velocity is hazardous. It is currently impossible to tell whether breccias or cataclasites result from slip at seismic strain rates in the absence of pseudotachylytes. I suggest a more careful writing.**

We agree with reviewer. However, ancient seismicity along the BFZ is documented by widespread occurrence of pseudotachylytes and, as documented in this work, microstructures typical of high-stress conditions (i.e., wall-rock quartz deformation lamellae and wall-rock pulverization) associated with the described sheared fault-veins. See detailed reply below (pages 11-13).

**Detailed points:**

**L.14: fault-veins / epidote faulting: what do you mean? Please clarify. Not clear enough for an abstract.**

With fault-veins, we mean hybrid-shear veins. We made these aspects clearer in the abstract. See reply to the next point.

**L.17-21: the abstract lacks material that enables understanding how you came to these conclusions.**

We rephrased and included material to the abstract to make it clearer.

Old: "*Earthquake swarms commonly occur in upper-crustal hydrothermal-magmatic systems and activate mesh-like fault networks. How these networks develop through space and time along seismic faults is poorly constrained in the geological record. Here, we describe a spatially dense array of small-displacement (< 1.5 m) epidote-rich fault-veins within granitoids, occurring at the intersections of subsidiary faults with the exhumed seismogenic Bolfin Fault Zone (Atacama Fault System, Northern Chile). Epidote faulting and veining occurred at 3-7 km depth and 200-300 °C ambient temperature. At distance ≤ 1 cm to fault-veins, the magmatic quartz of the wall-rock shows (i) thin (< 10-μm-thick) interlaced deformation lamellae, and (ii) crosscutting quartz-filled veinlets. The epidote-rich fault-veins (i) include clasts of deformed magmatic quartz, with deformation lamellae and quartz-filled veinlets, and (ii) record cyclic events of extensional-to-hybrid veining and either aseismic or seismic shearing. Deformation of the wall-rock quartz is interpreted to record the large stress perturbations associated with the rupture propagation of small earthquakes. In contrast, dilation and shearing forming the epidote-rich fault-veins are interpreted to record the later development of a mature and hydraulically-connected fault-fracture system. In this latter stage, the fault-fracture system cyclically ruptured due to fluid pressure fluctuations, possibly correlated with swarm-like earthquake sequences.*"

New: "*Earthquake swarms commonly occur in upper-crustal hydrothermal-magmatic systems and activate mesh-like fault networks. How these networks develop through space and time along seismic faults is poorly constrained in the geological record. Here, we describe a spatially dense array of small-displacement (< 1.5 m) epidote-rich fault-veins (i.e. hybrid extensional-shear veins) within granitoids, occurring at the intersections of subsidiary faults with the exhumed seismogenic Bolfin Fault Zone*

*(Atacama Fault System, Northern Chile). Epidote hybrid extensional-shear veining occurred at 3-7 km depth and 200-300 °C ambient temperature. At distance ≤ 1 cm to fault-veins, the magmatic quartz of the wall-rock shows (i) thin (< 10-µm-thick) interlaced deformation lamellae, and (ii) systematically crosscutting veinlets healed by quartz and feldspars, and appears shattered at the vein contact. Clasts of deformed magmatic quartz, with deformation lamellae and healed veinlets, are included in the epidote-rich fault-veins. Deformation of the wall-rock quartz is interpreted to record the transient large stress perturbation associated with the propagation of small earthquakes, preceding conspicuous epidote mineralization. Conversely, the epidote-rich fault-veins record cyclic events of extensional-to-hybrid veining and either aseismic or seismic shearing. The dilation and shearing behavior of the epidote-rich fault-veins are interpreted to record the later development of a mature and hydraulically-connected fault-fracture system. In this latter stage, the fault-fracture system cyclically ruptured due to fluid pressure fluctuations, possibly correlated with swarm-like earthquake sequences.*"

**L.65: linkage zone? What do you mean?**

We mean "zone of fault linkage". We modified it in the next version of the manuscript.

**L.129: replace by 'EBSD data was processed using the MTEX…'**

We modified the text as suggested by the reviewer.

**L.132: why such a low (6 nA) current?**

Because these working conditions were the best to get detailed resolute images.

**L.135: you mean alkali devolatilization?  (instead of migration)**

Yes, we modified the text in the revised version of our manuscript.

**L.202: (Al-rich; light: Fe-rich) there is something missing here**

(dark: Al-rich; light: Fe-rich).

**L.211: show the pores!**

Pores are shown in Fig. 6d, 6f and 6g. See revised figure 6 at pages 19-20.

**L.216: there is no figure 5h! check again the figure calls. Do you mean 6h?**

Yes. We thank the review for noting this typo. Correction done.

**L.267: 'high stresses': how high?**

Trepmann and Stöckhert (2013) reached confined pressure up to 2.7 GPa in their experiments.

**L.269: how slow?**

We mean strain rate slower than what expected for co-seismic deformation.

**L.282: 'the latter hypothesis': clarify to which hypothesis you are referring to**

We refer to the hypothesis that most quartz deformation lamellae we observed are possibly related to high-stress conditions rather than slow low-temperature crystal plasticity. We modified the text considering the comments by the reviewers. See detailed reply at pages 11-13.

**L.295-298: It is not clear to me how high stresses can be reached here in hybrid veins, since mode ii veins by definition require higher fluid pressures than pure shear veins**

Indeed, we refer to the high-stress perturbation at the crack tip propagating at seismic speeds. We clarified this in the revised version.

Old: "*Thus, in the relatively small-displacement (< 1.5 m) and up to 10s-m-long faults and hybrid fractures of the epidote-rich fault-vein networks, we interpret the occurrence of deformation lamellae in the wall-rock quartz to reflect the high-stress field associated with rupture tip propagation at seismic speeds during initial fracturing*"

New: "*Thus, in the relatively small-displacement (< 1.5 m) and up to 10s-m-long faults and hybrid fractures of the epidote-rich fault-vein networks, we interpret the occurrence of deformation lamellae in the wall-rock quartz to reflect the transient high-stress pulse associated with rupture tip propagation at seismic speeds during initial fracturing*"

**L.305: rephrase: too long and not clear**

We rephrased the sentence to make it clearer.

Old: "*Quartz-healed veinlets sharply crosscutting the quartz deformation lamellae (Figs. 3d-e, 4d, 4f) within the micro-damage zone of the epidote-rich fault-veins (Figs. 3a-c, 4) increase in spatial density towards the vein boundary (Fig. 4), are mostly oriented at high angle with respect to the vein boundary (Figs. 3d, 4f, 4h), and are healed by the minerals (quartz, K-feldspar and albite) of the crosscut wall-rock (Figs. 4c-h, 5).*"

New: "*The veinlets sharply and systematically crosscut the quartz deformation lamellae (Figs. 3-4, S1-S2), increase in spatial density towards the vein boundary (Fig. 4), are mostly oriented at high angle with respect to the vein boundary (Figs. 3d, 4h, 4j, S2), and are healed by the minerals (quartz and K-feldspar, K-feldspar and albite across quartz and feldspar grains, respectively) of the crosscut wall-rock (Figs. 4c-j, 5, S1-S2).*"

**L.314-317: Yes, but this study deals with dissolution precipitation of the host metasediments towards the host. In this study you suggest permeability increments associated with fluid advection.**

We agree with the reviewer but a previous reviewer suggested to take in consideration these works and we discuss them.

**L.326: any evidence for pressure-solution at this stage that could account for elemental redistribution?**

No, we did not observe any evidence of pressure-solution in the wall-rocks.

**L.351: see Angiboust et al. (2015, G-cubed) for another example where an epidote-rich cataclazed fault system is described, forming as a consequence of transients increase in pore fluid pressure in a sheared system. See also Oncken et al. (2021, geosphere) for further evidence of foliated cataclasites as a key fault zone material in deep plate boundary systems. See also Muñoz-Montecinos et al. (EPSL, 2021).**

We took in consideration these papers in the revised version of our manuscript and cited them. See detailed reply to the next point.

**L.353: foliated, fluidized cataclasites and breccias may also form at sub-seismic strain rates (see Oncken et al., 2021). I would be more careful in the writing here.**

We thank the reviewer for their suggestion, but the foliated and fluidized cataclasites described by Oncken et al. (2022) are associated with tiny pseudotachylytes. Thus, this example is different from the foliated cataclasites. We think that it is more appropriated to refer to the examples described by Angiboust et al. (2015) and Muñoz-Montecinos et al. (2021) for cataclasites associated with aseismic creep and possible seismic slip, respectively.

Old: "*Some cataclasites are foliated (Fig. 6e) suggesting that slip likely occurred by aseismic fault creep (e.g., Chester and Chester, 1998; Rutter et al., 1986). On the other hand, most cataclasites display suspended clasts of wall-rocks and earlier veins (Figs. 2d, 3a, 6a, 6c, 6h-j) similar to the microstructures observed in fluidized cataclasites and breccias, which have been interpreted as markers of co-seismic slip (e.g., Berger and Herwegh, 2019; Cox and Munroe, 2016; Fondriest et al., 2012; Masoch et al., 2019; Smith et al., 2008).*"

New: "*Some cataclasites are foliated (Fig. 6e) suggesting that slip likely occurred by aseismic fault creep (e.g., Angiboust et al., 2015; Chester and Chester, 1998; Rutter et al., 1986). On the other hand, most cataclasites display suspended clasts of wall-rocks and earlier veins (Figs. 2d, 3a, 6a, 6c, 6h-j) similar to the microstructures observed in fluidized cataclasites and breccias, which have been interpreted as markers of co-seismic slip (e.g., Berger and Herwegh, 2019; Cox and Munroe, 2016; Fondriest et al., 2012; Masoch et al., 2019; Muñoz-Montecinos et al., 2021; Smith et al., 2008).*"

**L.370: check syntax in this sentence (perhaps you mean 'by an increase of the rate of fluid pressure…')**

Yes, thanks for noting this misleading sentence. We modified the text and Figure 9 also considering the observation of the reviewer about fault reactivation (point below). We modified the text in the revised version of our manuscript as below.

Old: "*The described deformation cycle can repeatedly occur if the system is dominated by increase rate of fluid pressure larger than increase rate of tectonic loading (Cox, 2016; Phillips, 1972)*"

New: "*The described deformation cycle can repeatedly occur if the system is dominated by an increase in the rate of fluid pressure larger than an increase in the rate of tectonic loading (Cox, 2016; Phillips,*

1972). Moreover, as in the studied samples, extensional veining precedes either hybrid and shear failure, it is likely that epidote and prehnite sealing promotes recovery of cohesive strength in the timescales of rupture recurrence during a swarm (Fig. 9)."

**L.372: which type of deformation events?**

Independent events of extensional faulting followed by fault reactivation.

**L.384: coexist (no 's')**

Yes, thanks for noting this typo. Correction done.

**L.430: the presence of suspended clasts within cataclasite should not be viewed alone as a solid evidence for seismic slip**

We agree with the reviewer. Indeed, we specified that this microstructure is a possible marker of paleo-earthquake.

**Bird & Spieler (2004, rev. Min. Geochem) could be a useful reference when it comes to demonstrating that your veins were once part of a supra-plutonic environment.**

See response above about fluid sources (pages 21-22).

**Figure 1: how do you define the 'weakly fractured unit'? is there a statistical criterion or is it purely arbitrary? Also, explain better your difference between chloritized/Fractured and chlorite-rich cataclastic: foliated and massive, in the caption of the map.**

The structural units forming the BFZ are presented in detail in Masoch et al. (2022) and their classification was based on a quantitative analysis. In that work, we defined the different units based on the following structural features: preservation of original magmatic features of the host rocks, alteration degree of the host rocks, spacing of fractures, veins and faults, relatively abundance of veins and faults, mineral assemblage sealing and decorating veins and faults, and clast/matrix proportion in the fault rocks.

**Figure 3: better label minerals and features as explained in the caption**

We improved the labelling. See revised figure 3 above.

**Figure 4: what are the analytical conditions used for obtaining these CL images?**

The working conditions at which the CL images were acquired are described in the methods section of our manuscript.

**Figure 5b: how many points considered for this pole figure? figure 5a: why not showing the misorientation map instead? It would be more useful here. What is the Y direction and what is the reference frame?**

We plotted 833 points in the pole figure. Fig. 5a shows the orientation of the deformation lamellae compared to the crystallographic orientation of the host quartz grain. Microstructural observations were conducted parallel to the fault lineation (X direction) and orthogonal to the fault/vein wall (X-Y plane), as specified in the methods section.

**Figure 8: part (a) title: propagation (not progation).  The three lines (crystal plasticity, dynamic fracturing …) are not well positioned and not very easy to understand in the frame of this sketch.**

We modified the figure as below hoping now it is clearer. We thank the reviewer for highlighting the typo.

[Figure]

**Figure 9: ok but this figure does not consider the reactivation of the fault**

We modified the figure adding the curve of re-shear for incohesive rocks. We implemented also the text about fault reactivation (see point above).

[Figure]

**References cited in the rebuttal letter**

Angiboust, S., Kirsch, J., Oncken, O., Glodny, J., Monié, P., Rybacki, E., 2015. Probing the transition between seismically coupled and decoupled segments along an ancient subduction interface. Geochemistry, Geophys. Geosystems 16, 1905–1922. https://doi.org/10.1002/2015GC005776

Masoch, S., 2023. Structure, evolution and deformation mechanisms of crustal-scale seismogenic faults (Bolfin Fault Zone, Northern Chile). PhD thesis. Università degli Studi di Padova.

Masoch, S., Fondriest, M., Gomila, R., Jensen, E., Mitchell, T.M., Cembrano, J., Pennacchioni, G., Di Toro, G., 2022. Along-strike architectural variability of an exhumed crustal-scale seismogenic fault (Bolfin Fault Zone, Atacama Fault System, Chile). J. Struct. Geol. 165, 104745. https://doi.org/10.1016/j.jsg.2022.104745

Muñoz-Montecinos, J., Angiboust, S., Garcia-Casco, A., 2021. Blueschist-facies paleo-earthquakes in a serpentinite channel (Zagros suture, Iran) enlighten seismogenesis in Mariana-type subduction margins. Earth Planet. Sci. Lett. 573, 117135. https://doi.org/10.1016/j.epsl.2021.117135

Oncken, O., Angiboust, S., Dresen, G., 2022. Slow slip in subduction zones: Reconciling deformation fabrics with instrumental observations and laboratory results. Geosphere 18, 104–129. https://doi.org/10.1130/GES02382.1

Trepmann, C.A., Stöckhert, B., 2013. Short-wavelength undulatory extinction in quartz recording coseismic deformation in the middle crust – an experimental study. Solid Earth 4, 263–276. https://doi.org/10.5194/se-4-263-2013